# MEK inhibitors activate Wnt signalling and induce stem cell plasticity in colorectal cancer

Tianzuo Zhan[1,2,7], Giulia Ambrosi[1,7], Anna Maxi Wandmacher[1,7], Benedikt Rauscher[1], Johannes Betge [1,2], Niklas Rindtorff[1], Ragna S. Häussler[3], Isabel Hinsenkamp[2], Leonhard Bamberg[2], Bernd Hessling[4], Karin Müller-Decker[5], Gerrit Erdmann [6], Elke Burgermeister[2], Matthias P. Ebert[2] & Michael Boutros [1]

In colorectal cancer (CRC), aberrant Wnt signalling is essential for tumorigenesis and maintenance of cancer stem cells. However, how other oncogenic pathways converge on Wnt signalling to modulate stem cell homeostasis in CRC currently remains poorly understood. Using large-scale compound screens in CRC, we identify MEK1/2 inhibitors as potent activators of Wnt/β-catenin signalling. Targeting MEK increases Wnt activity in different CRC cell lines and murine intestine in vivo. Truncating mutations of APC generated by CRISPR/Cas9 strongly synergize with MEK inhibitors in enhancing Wnt responses in isogenic CRC models. Mechanistically, we demonstrate that MEK inhibition induces a rapid downregulation of AXIN1. Using patient-derived CRC organoids, we show that MEK inhibition leads to increased Wnt activity, elevated *LGR5* levels and enrichment of gene signatures associated with stemness and cancer relapse. Our study demonstrates that clinically used MEK inhibitors inadvertently induce stem cell plasticity, revealing an unknown side effect of RAS pathway inhibition.

[1] Division Signaling and Functional Genomics, German Cancer Research Center (DKFZ) and Heidelberg University, 69120 Heidelberg, Germany.
[2] Department of Internal Medicine II of the Medical Faculty Mannheim, Heidelberg University, 68167 Mannheim, Germany. [3] NMI Natural and Medical Sciences Institute at the University of Tübingen, 72770 Reutlingen, Germany. [4] Proteomics Core Facility, German Cancer Research Center (DKFZ), 69120 Heidelberg, Germany. [5] Core Facility Tumor Models, German Cancer Research Center (DKFZ), 69120 Heidelberg, Germany. [6] NMI TT Pharmaservices, 13353 Berlin, Germany. [7] These authors contributed equally: Tianzuo Zhan, Giulia Ambrosi, Maxi Anna Wandmacher. Correspondence and requests for materials should be addressed to M.B. (email: m.boutros@dkfz.de)

 1

Colorectal cancer (CRC) is one of the major cancers with a high burden on society[1]. While treatment options of CRC have expanded over the last decades, the survival rates of advanced stage cancers remain poor[1]. To understand the biology of CRC, genome sequencing of many cancers was performed and revealed highly recurrent mutations in oncogenic signalling pathways[2]. Most genetic alterations were found in Wnt pathway components, underlining the importance of Wnt signalling for CRC[3]. Beyond the well-established function of Wnt in initiation of CRC, there is accumulating evidence supporting an outstanding role of Wnt signalling for the maintenance of stem cell homeostasis in both normal intestine and CRC[4]. In CRC stem cells, Wnt is particularly active and required for maintaining the stemness phenotype[5]. However, Wnt levels are heterogeneously distributed in CRC tissues and can dynamically fluctuate within a population of cancer cells, despite the presence of the same APC mutations[6]. These observations were recently integrated into a revised model of stem cell plasticity which proposes that cancer cells can dynamically shift between a differentiated and a stem-like state[7]. This stem cell plasticity is tightly linked to changes in Wnt levels, and factors that can modulate plasticity are gradually being revealed. For instance, genetic alteration of the Wnt pathway component LGR5[8,9] or the transcription factor HOXA5[10] can activate stem cell plasticity in CRC. Beyond these cell intrinsic signals, secreted cues from myofibroblasts in the microenvironment can also modulate Wnt activity and restore cancer stemness[6]. An important role of stem cell plasticity for cancer therapy is emerging, as cancer cells with stemness properties were found to be drivers of drug resistance in many cancers[11]. While most fast proliferating cells can be eradicated by chemotherapy, quiescent cancer stem cells are able to survive antineoplastic therapy, revert to fast proliferating cancer cells by cellular plasticity and serve as a clonogenic reserve for tumour regrowth[12,13]. To improve the treatment of CRC, it is therefore necessary to understand how other oncogenic pathways converge on Wnt signalling and modulate the plasticity of cancer stem cells.

Here, we describe activation of Wnt signalling and induction of stemness upon inhibition of the RAS pathway. We identified MEK inhibitors as potent activators of the canonical Wnt pathway in different CRC cell lines by high-throughput compound screens and confirmed this observation across many models of normal and transformed intestinal cells. Using isogenic cell lines generated by CRISPR/Cas9, we show that APC truncation can strongly enhance the Wnt activating effect of MEK1/2 inhibitors. Proteomic experiments identified a reduction of AXIN1 levels upon MEK1/2 inhibition and a dissociation of GSK3B from AXIN1. Finally, we provide evidence that a clinically used MEK inhibitor affects cellular plasticity and induces an intestinal stem cell signature in patient-derived cancer organoids that is predictive of disease relapse.

## Results

### Compound screens identify MEK inhibitors as Wnt activators.
Active Wnt signalling is critical for the progression of CRC and was found to confer resistance to chemotherapy and irradiation in different tumour types[14–16]. So far, only few small molecules targeting the Wnt pathway were identified, of which none has been approved for clinical use[3,17]. To discover small molecules that modulate Wnt signalling in CRC, we performed several compound screens in three CRC lines (HCT116, SW480 and DLD1) (Fig. 1a). The selected cell lines harbour common mutations of KRAS (G12V in SW480 and G13D in HCT116 and DLD1) and alterations of APC (DLD1, SW480) and β-catenin (HCT116). To generate stable Wnt reporter cell lines, we infected cancer cells with lentivirus encoding TCF-Wnt luciferase reporter

plasmids[18,19]. The responsiveness of the reporter cell lines to perturbations of the Wnt pathway was confirmed by treatments with the GSK3 inhibitors BIO and CHIR99021, the tankyrase inhibitor IWR-1, the CSNK1A1 activator pyrvinium and the porcupine inhibitor LGK974 (Supplementary Fig. 1A).

We then used a small molecule library comprising 2399 compounds to discover modulators of Wnt signalling (see Methods and Supplementary Data 1). CRC cell lines were treated with small molecules for 24 h, followed by cell lysis and parallel measurement of Wnt reporter activity and cell viability. In line with our pre-test results, we found that BIO increased Wnt reporter activity, whereas IWR-1 and LGK974 mildly reduced it (Fig. 1b). Surprisingly, among the candidate compounds that modified Wnt activity, we discovered that the MEK1/2 inhibitor PD-0325901 increased Wnt reporter levels across all three CRC lines (see Supplementary Data 1 and Supplementary Fig. 1B).

To test whether the Wnt activating effect is specific for the class of MEK1/2 inhibitors, we performed secondary screens using a focused kinase inhibitor library. This library consisted of 273 compounds targeting a diverse set of kinases, including many FDA approved drugs directed against the RAS pathway (see Supplementary Data 2). We found that almost all MEK1/2 inhibitors activated Wnt signalling, including the well characterised small molecules trametinib, selumetinib, U0126 and PD318088. Interestingly, the mean effect size of MEK1/2 inhibitors on Wnt activity was not significantly different compared to GSK3 inhibitors (Fig. 1c, Supplementary Fig. 2A). In contrast, small molecules targeting EGFR or BRAF/RAF1 did not increase Wnt reporter levels (Fig. 1c–d) (see Supplementary Data 2). The ability to activate Wnt signalling was shared among MEK1/2 inhibitors that have divergent chemical structures, suggesting that Wnt activation is caused by an on-target effect (Fig. 1e). Measurement of phospho-ERK and phospho-RSK levels showed that only pharmacological inhibition of MEK, but not of EGFR or RAF could abolish downstream RAS signalling (Supplementary Fig. 2B). This result indicates that sufficient blockage of the pathway at the level of ERK is necessary for activation of Wnt. Taken together, these cell-based screening experiments identified MEK1/2 inhibitors as potent activators of Wnt signalling.

### Trametinib is a potent activator of canonical Wnt signalling.
To validate the results of the screen, we selected the MEK1/2 inhibitors selumetinib, trametinib and PD318088 for further experiments. As shown in Fig. 2a–b, all compounds induced a dose-dependent increase of TCF-Wnt reporter levels and expression of AXIN2 in SW480. Since trametinib is an FDA approved MEK1/2 inhibitor and currently tested in clinical trials as an antineoplastic agent for the treatment of CRC (e.g. NCT03087071, NCT03377361), we focused on this compound. First, in dose-response experiments, we showed that activation of Wnt occurs at low concentrations of trametinib (10 nM) and reaches a plateau at 100 nM (Fig. 2c). Furthermore, MEK1/2-induced expression of the Wnt target gene AXIN2 occured in a time-dependent manner. Activation of Wnt started 4 h after addition of trametinib and increased with incubation time (Fig. 2d). Furthermore, trametinib stimulated expression of AXIN2 across KRAS (HCT116, SW403) and BRAF (HT29) mutant CRC cell lines (Fig. 2e).

To test whether Wnt target gene expression can also be induced by RNAi mediated knockdown of MEK1/2, we transfected HCT116 with siRNA against MEK1 or MEK2. Single knockdown of either gene did not alter AXIN2 expression, but combined knockdown of MEK1 and MEK2—which resembles the effect of pharmacological MEK inhibitors—significantly increased expression of AXIN2 (Fig. 3a, Supplementary Fig. 3). Interestingly, we found that single

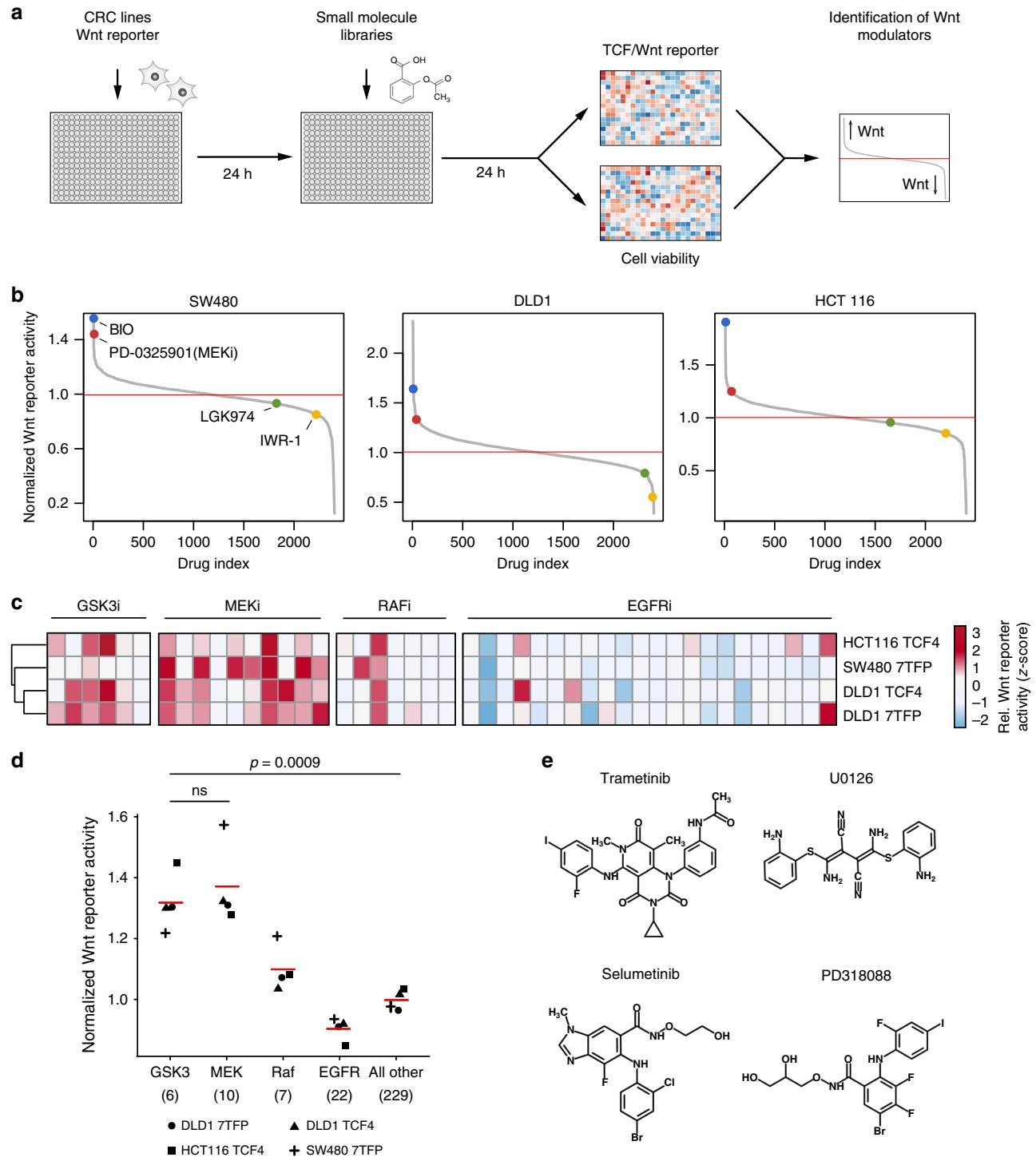

and combined knockdown of ERK1 and ERK2 also induced *AXIN2* transcript levels (Fig. 3b). This observation could be confirmed by two independent pharmacological inhibitors of ERK1/2, GDC-0994 and SCH772984, which increased *AXIN2* transcript and protein levels in both HCT116 and SW480 (Fig. 3c–e).

Next, we asked whether MEK1/2 dependent activation of Wnt activity requires the canonical Wnt pathway. We used siRNAs to knockdown expression of β-catenin (CTNNB1) in HCT116, which resulted in a strong reduction of *AXIN2* levels. Treatment with trametinib did not increase *AXIN2* levels when β-catenin was knocked down in parallel (Supplementary Fig. 4A), indicating that β-catenin is required and acts downstream of the MEK1/2-

induced Wnt activation. Similarly, co-treatment of HCT116 with ICG-001, an inhibitor of β-catenin and CBP interaction, abolished the effect of trametinib on *AXIN2* expression (Supplementary Fig. 4B). These experiments suggest that trametinib induced Wnt activation targets a component upstream of β-catenin.

**Synergistic effect of trametinib and APC truncations.** We initially observed that the effect of trametinib on Wnt is divergent across individual CRC lines. For instance, treatment of RKO cells with trametinib resulted in only a minor activation of Wnt signalling. Unlike the other tested CRC lines, RKO cells do not

**Fig. 1** Compound screens identify MEK inhibitors as activators of Wnt in CRC. **a** Schematic overview of the screening procedure. Colorectal cancer cell lines stably expressing TCF/Wnt luciferase reporters were seeded onto 384 well plates in two separate sets. 24 h after seeding, compound libraries were added and cells were treated for 24 h, followed by measurement of cell viability in one set and Wnt reporter activity in the other set of plates. Two biological replicates were performed for each screen. **b** A large compound screen identifies PD-0325901 as an activator of Wnt signalling. Waterfall plots showing the effect of a compound library containing 2399 drugs on Wnt reporter activity in HCT116, SW480 and DLD1 cell lines. The MEK inhibitor PD-0325901 is shown as a red dot. The GSK3 inhibitor BIO serves as positive control, whereas the porcupine inhibitor LGK974 and tankyrase inhibitor IWR-1 are negative controls for Wnt reporter activity (see also Supplementary Fig. 1). The mean value of two independent experiments is presented. **c**–**d** A kinase-focused compound screen confirms Wnt activation by MEK inhibitors. Four colorectal cancer cell lines stably expressing TCF/Wnt reporters were treated with a compound library containing 274 kinase inhibitors. **c** Heatmap of Wnt reporter activities for all EGFR, RAF, MEK and GSK3 inhibitors. Wnt reporter activity was z-normalised for all drugs and high activity is presented in red and low activity in blue. **d** Dot plot showing average Wnt reporter activity levels resulting from treatment with different classes of Ras pathway and GSK3 inhibitors. Wnt reporter activity is significantly increased by MEK inhibitors compared to all other kinase inhibitors, and the increase is similar to GSK3 inhibition (two-sided Student's t-test). Mean values of two independent experiments are presented. **e** Chemical structure of selected MEK1/2 inhibitors trametinib (PubChem CID 11707110), U0126 (Pubchem CID 3006531), selumetinib (PubChem CID 10127622) and PD-318088 (PubChem CID 10231331)

harbour mutations in any components of the destruction complex. Instead, the E3 ubiquitin-ligase *RNF43* is mutated, resulting in a sensitization of Wnt signalling at the receptor level[20]. We hypothesised that alteration of *APC* could potentially enhance the effect of trametinib on Wnt signalling in this cell line. To address this question, we genetically engineered a truncation of the *APC* gene in RKO using CRISPR/Cas9. We selected an sgRNA targeting a region of the gene that frequently harbours loss-of-function mutations (Fig. 4a) and generated two isogenic clones with distinct homozygous APC truncations. Correct genome editing of *APC* was verified by amplicon sequencing and by Western Blot (Fig. 4b–c). APC truncation resulted in a higher basal level of Wnt activity in edited cells (Fig. 4d–e). Unlike the wild type, the APC truncated cell lines responded to trametinib treatment with a strongly enhanced Wnt activation, demonstrated by both *AXIN2* expression and Wnt reporter levels (Fig. 4d). This observation was consistent for both genetically modified clones of RKO. Similarly, treatment with the ERK1/2 inhibitor GDC-0994 resulted in an increase of Wnt signalling in RKO with truncated, but not wild-type APC (Fig. 4e). Compared to trametinib, ERK inhibition was less potent in activating *AXIN2* expression, but increased Wnt reporter activity to a similar degree (Fig. 4d–e). Interestingly, GSK3 inhibition was unable to induce Wnt activity in RKO cells after APC truncation, indicating that MEK/ERK and GSK3 inhibition activate Wnt by distinct mechanisms. These results show that the presence of *APC* mutations can strongly enhance the Wnt stimulating effect of MEK1/2 inhibitors.

**AXIN1 is regulated by MEK inhibition**. To investigate whether MEK1/2 inhibition affects the protein levels of key Wnt pathway components, we performed a bead-based Western Blot (DigiWest) experiment that enabled the parallel measurement of hundreds of proteins[21]. We selected 128 antibodies against components of the Wnt and RAS pathway and measured abundancies of both proteins and phosphoproteins (Supplementary Table 1). HCT116 cells were treated with trametinib for 24 h, lysed and then analysed by DigiWest. As shown in Fig. 5a, phospho-ERK and its downstream target phospho-RSK were reduced in trametinib treated cells, demonstrating effective inhibition of the RAS pathway. At the same time, protein levels of the Wnt target genes *AXIN2*, *NKD1* and of *TAZ* were slightly upregulated, corresponding to an activation of the Wnt pathway. Interestingly, we observed a reduction in AXIN1 protein levels upon trametinib treatment, which was consistent across all biological replicates (Fig. 5a, Supplementary Fig. 5).

Since AXIN1 is considered to be a rate-limiting factor of the destruction complex[22], we sought to further investigate its role in

MEK1/2 inhibitor induced Wnt activation. First, we confirmed the findings of DigiWest by Western Blot analysis in HCT116 and SW480, and showed that AXIN1 protein levels were decreased upon trametinib treatment in both cell lines (Fig. 5b, Supplementary Fig. 6A). To quantify the loss of AXIN1, we applied an independent proteomics approach. AXIN1 proteins from trametinib treated HCT116 were affinity purified and the protein abundance was measured by mass spectrometry. Upon treatment, we observed on average a 50-fold decrease in AXIN1 proteins levels compared to DMSO treated controls (Supplementary Fig. 6B). To investigate the dynamics of AXIN1 reduction in the presence of MEK inhibition, we measured proteins levels within short time periods after addition of trametinib (1, 2 and 4 h). Furthermore, we inhibited de novo protein synthesis using cycloheximide. As shown in Fig. 5c, cytoplasmic levels of AXIN1 started to decrease after 2 h of trametinib treatment. This effect was even more pronounced in cells treated with cycloheximide. Simultaneous measurement of cellular AXIN1 and AXIN2 transcript levels showed that loss of AXIN1 protein is accompanied by an induction of *AXIN2* and reduction of *AXIN1* transcript levels (Fig. 5d). This decrease of *AXIN1* expression by ~50% was maintained 24 h after incubation with trametinib (Supplementary Fig. 6C).

To assess the importance of AXIN1 levels in regulating canonical Wnt activity, we first performed siRNA mediated knockdown of *AXIN1* in HCT116. Using three functional siRNAs, we showed that the expression of the Wnt target gene *AXIN2* is strongly enhanced upon knockdown of *AXIN1*, which was comparable to the effect of siRNAs targeting *APC* (Fig. 5e). In addition, we demonstrated that pharmacological inhibition of ERK could reduce AXIN1 protein levels, albeit to a lesser extent than MEK inhibitors (Supplementary Fig. 6D). Next, we sought to understand if stabilizing AXIN1 levels can rescue the effect of MEK1/2 inhibitors on Wnt signalling. For this purpose, we performed co-treatment experiments with the tankyrase inhibitor XAV939 and trametinib. Inhibition of tankyrases by XAV939 prevents degradation of AXIN1, leading to reduced Wnt activity[23]. In HCT116, the stabilizing effect of XAV939 on AXIN1 overrode the reduction of AXIN1 levels caused by trametinib (Supplementary Fig. 6E). As a result, trametinib was unable to induce *AXIN2* expression in the presence of XAV939. Thus, we demonstrated that fine-tuning of AXIN1 levels can strongly influence Wnt signalling and is a critical mechanism by which trametinib activates the Wnt pathway.

Next, we sought to identify mechanistic links between RAS signalling and transcriptional regulation of *AXIN1*. In the DigiWest experiment, we discovered that protein levels of the transcription factor *EGR1* were strongly depleted upon trametinib treatment (Supplementary Fig. 7A-B). *EGR1* is a known

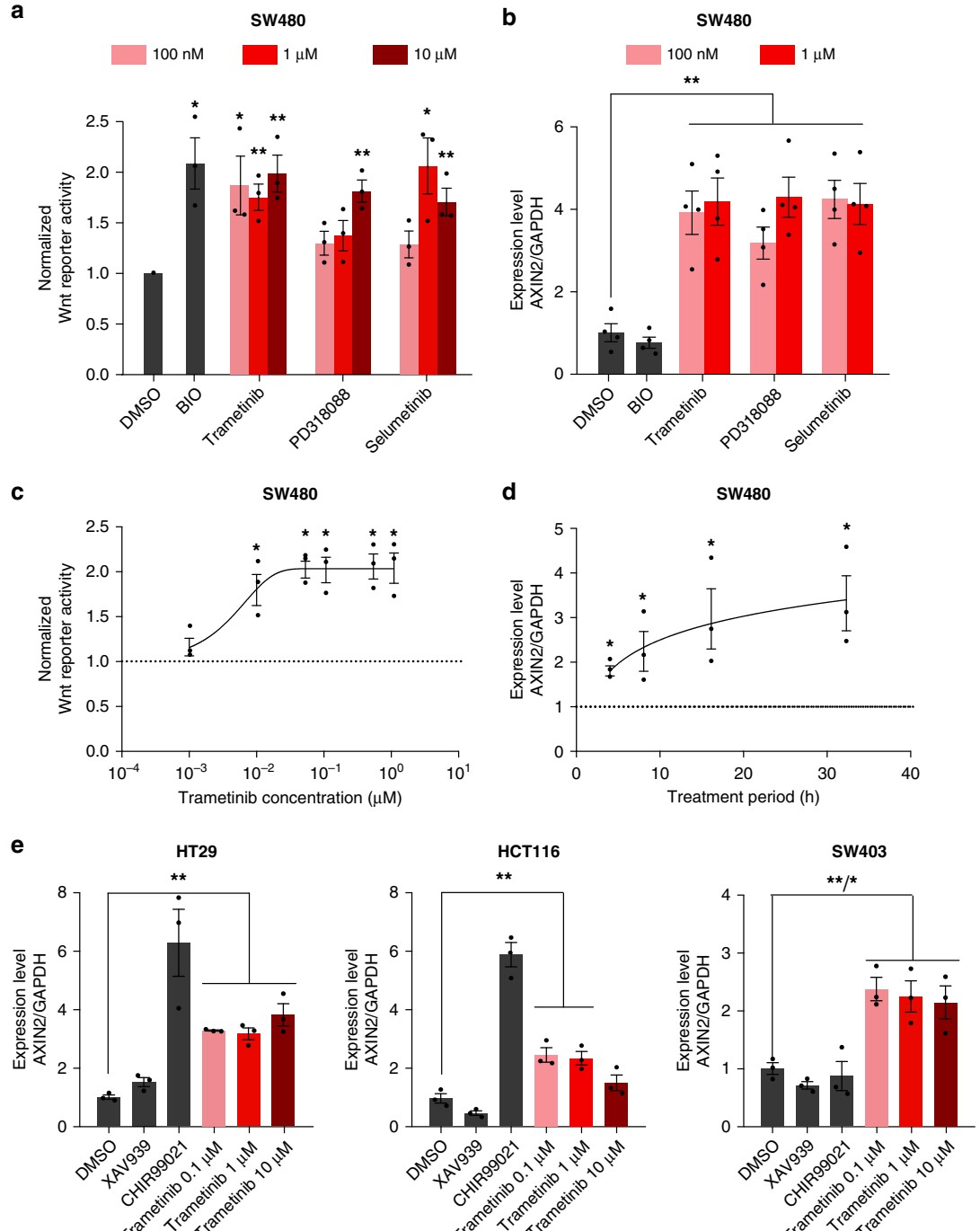

**Fig. 2** Characteristics of Wnt activation by MEK inhibitors. **a–b** Wnt reporter activity and target gene expression are increased by MEK inhibitors. SW480–7TFP were treated with different concentrations of the MEK inhibitors selumetinib, PD318088 or trametinib. TCF/Wnt-luciferase and CellTiterGlo signals were determined and normalised to DMSO controls (**a**) and expression of *AXIN2* was measured by qPCR (**b**). **c** Concentration dependent activation of Wnt signalling by MEK inhibitor trametinib. SW480–7TFP cells were treated with different concentrations of trametinib for 24 h. TCF/Wnt reporter activity and CellTiterGlo signal were determined and normalised to DMSO controls. **d** Time-dependent activation of Wnt signalling by trametinib. SW480 cells were treated for different time periods with 1 µM of trametinib and expression of Wnt target gene *AXIN2* was measured by qPCR. **e** Trametinib activates Wnt signalling in different colorectal cancer cell lines. HT29, HCT116 and SW403 cells were treated with indicated concentrations of trametinib for 24 h and *AXIN2* transcript levels were determined by qPCR. The tankyrase inhibitor XAV939 serves as negative and the GSK3 inhibitor CHIR99021 as positive control. **a–e** Data from three independent experiments are presented as mean ± s.e.m. *$p < 0.05$, **$p < 0.01$, two-sided Student's *t*-test

downstream target of the RAS pathway[24,25] and binds to the enhancer region of *AXIN1*, thereby activating its expression[26]. We measured the dynamics of *EGR1* loss in HCT116 and found that protein and transcript levels are both rapidly decreased after 1–2 h of treatment with trametinib (Supplementary Fig. 7C),

suggesting a process that occurs in parallel to the reduction of AXIN1 levels. We then investigated whether the effect observed in cell lines can also be confirmed in CRC organoids. For this purpose, four different organoid lines were treated with trametinib and *EGR1* transcript levels were determined by qPCR

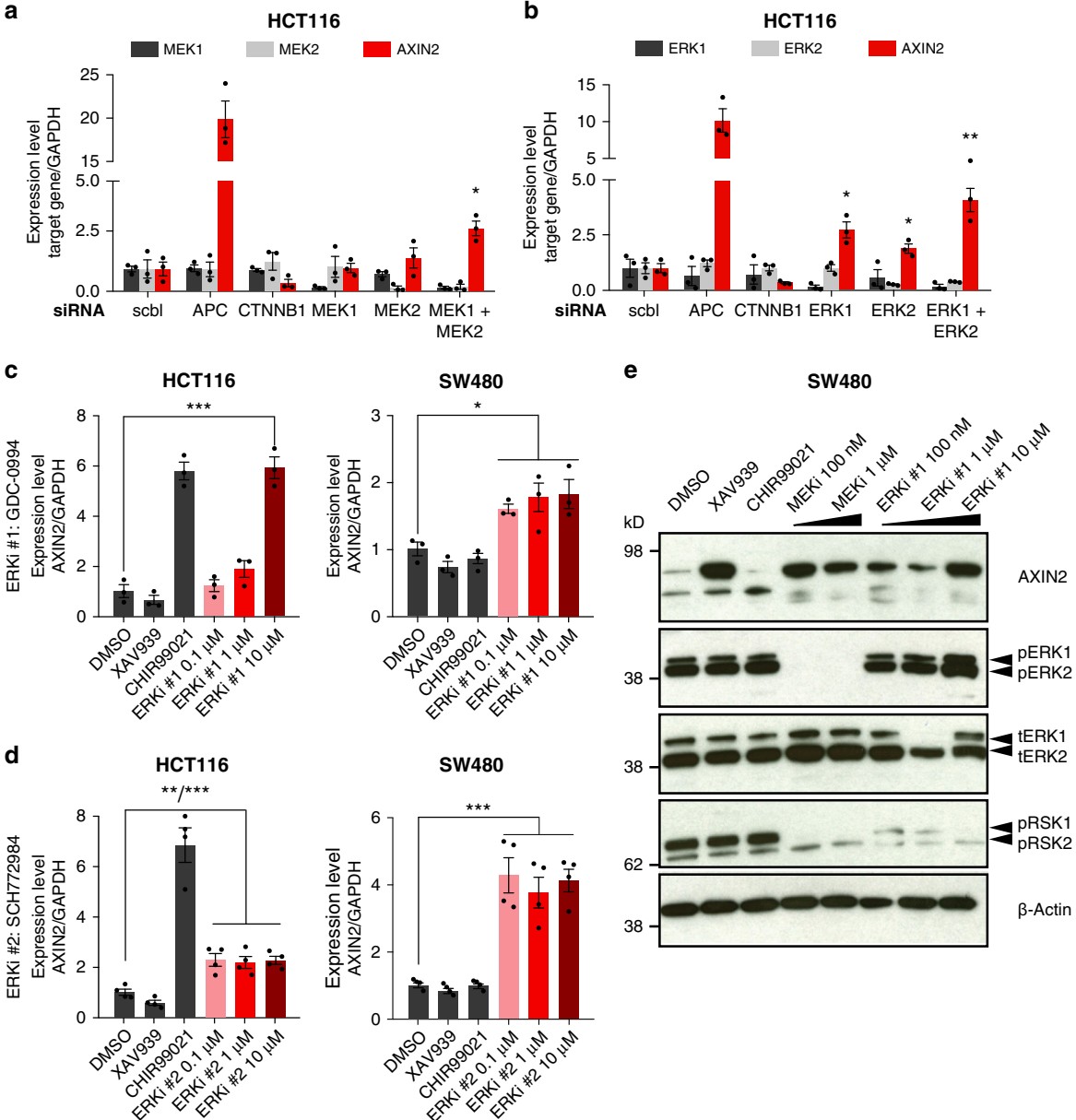

**Fig. 3** Inhibition of MEK and ERK by RNAi and small molecules activates canonical Wnt. **a–b** siRNA mediated knockdown of MEK1/2 and ERK1/2 activates Wnt signalling. HCT116 cells were transfected with the indicated siRNAs for 72 h, followed by measurement of expression levels of target genes by qPCR. The efficiency of individual siRNAs is shown in Supplementary Fig. 2. Data from three independent experiments are presented as mean ± s.e.m. scbl: scrambled siRNA. **c–e** Small molecule inhibitors of ERK1/2, GDC-0994 and SCH772984, activate the Wnt pathway. HCT116 and SW480 cells were treated with different concentrations of GDC-0094 (ERKi #1) (**c**) or SCH772984 (ERKi #2) (**d**) for 24 h. Transcript levels of *AXIN2* were measured by qPCR and data from three or four independent experiments are presented as mean ± s.e.m. Protein levels of AXIN2 and of RAS-MAPK cascade components ERK and RSK were determined by Western Blot in trametinib treated SW480 cells (**d**). A representative Western Blot of three independent replicates is shown. Dependence of MEK inhibitor induced Wnt activation on β-catenin is shown in Supplementary Fig. 4. *$p < 0.05$, **$p < 0.01$, ***$p < 0.001$, two-sided Student's *t*-test

(Supplementary Fig. 7D). In all lines, EGR1 expression was strongly repressed upon MEK inhibition. Finally, we analysed if transient overexpression of EGR1 could rescue downregulation of *AXIN1* expression by MEK inhibitors. HCT116 were either mock-transfected or transfected with an expression plasmid encoding mCherry or *EGR1*, which resulted in an overexpression of the respective genes. Upon treatment with trametinib, both *EGR1* and *AXIN1* levels decreased in the mock-transfected controls and mCherry expressing cells. In contrast, *EGR1* levels were only mildly reduced in cells overexpressing *EGR1*, with *AXIN1* transcript levels remaining unchanged (Supplementary

Fig. 7E). Therefore, our results suggest that *EGR1* could be a mechanistic link between RAS signalling and transcriptional regulation of *AXIN1* expression.

However, we also observed that the relative effect of AXIN1 depletion on Wnt activity is divergent between cell lines. For instance, we generated SW480 lines with stable and efficient knockdown of *AXIN1* using CRISPRi. In these cell lines, we observed only a mild increase in *AXIN2* expression (Supplementary Fig. 6F). In previously performed mass spectrometry experiments, we found that binding partners of AXIN1 within the destruction complex can change upon MEK inhibition. To

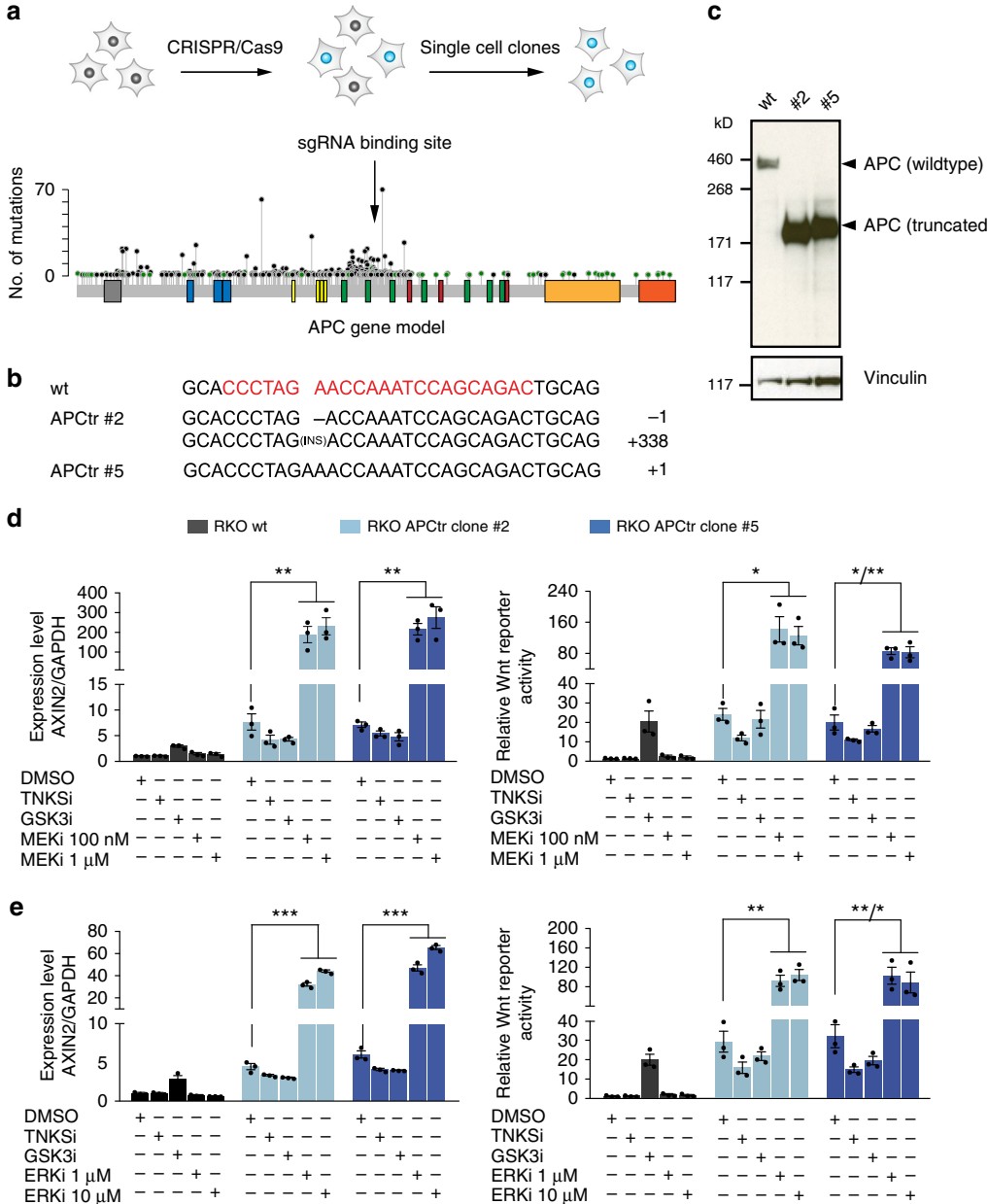

**Fig. 4** Synergistic effect of APC truncation and Ras pathway inhibition on Wnt activity. **a–c** Gene editing of APC truncations in RKO cells using CRISPR/Cas9. RKO cells harbouring wild-type *APC* were used for gene editing. A single sgRNA targeting the frequently mutated *CTNNB1* binding region of *APC* was cloned into the px459 vector, which encodes both Cas9 and a sgRNA expression cassette. RKO cells were then transiently transfected with the plasmid. After antibiotic selection, single cell clones were isolated and expanded (**a**). Correct genome editing was confirmed by targeted amplicon sequencing of the edited gene locus (**b**) and by Western Blot of APC (**c**). **d–e** Wnt signalling is strongly enhanced by MEK and ERK inhibition in an *APC* mutant background. Wild-type and two APC truncated RKO clones (APCtr) were treated for 24 h with either trametinib (MEKi) (**d**) or GDC-0994 (ERKi) (**e**). Wnt activity was determined via the expression of target gene *AXIN2* (left) and by TCF4/Wnt reporter activity (right). Data from three independent experiments are presented as mean ± s.e.m. *$p < 0.05$, **$p < 0.01$, ***$p < 0.001$, two-sided Student's *t*-test

further validate this observation, we performed affinity precipitation using antibodies against AXIN1 and GSK3B, and subsequent analysis of protein abundances of destruction complex members by Western Blot. In both affinity precipitations, a clear dissociation of GSK3B from AXIN1 was independently observed (Fig. 5f). The interaction of AXIN1 with GSK3B is critical in maintaining the integrity of the destruction complex[27] and dissociation of AXIN1 from GSK3B is known to occur during Wnt activation[28]. Thus, the observed loss of interaction between AXIN1 and GSK3B suggests an additional mechanism by which MEK1/2 inhibitors stimulate Wnt signalling.

In summary, we show that MEK1/2 inhibition results in two mechanisms that converge on AXIN1 and lead to activation of Wnt. These mechanisms are transcriptional downregulation of *AXIN1*, mediated by *EGR1*, and reshaping of the destruction complex by dissociation of AXIN1 from its binding partner GSK3B.

**MEK inhibition activates Wnt signalling in murine intestine.** Next, we investigated whether the Wnt activating effect of trametinib in cancer cell lines can also be observed in vivo in murine

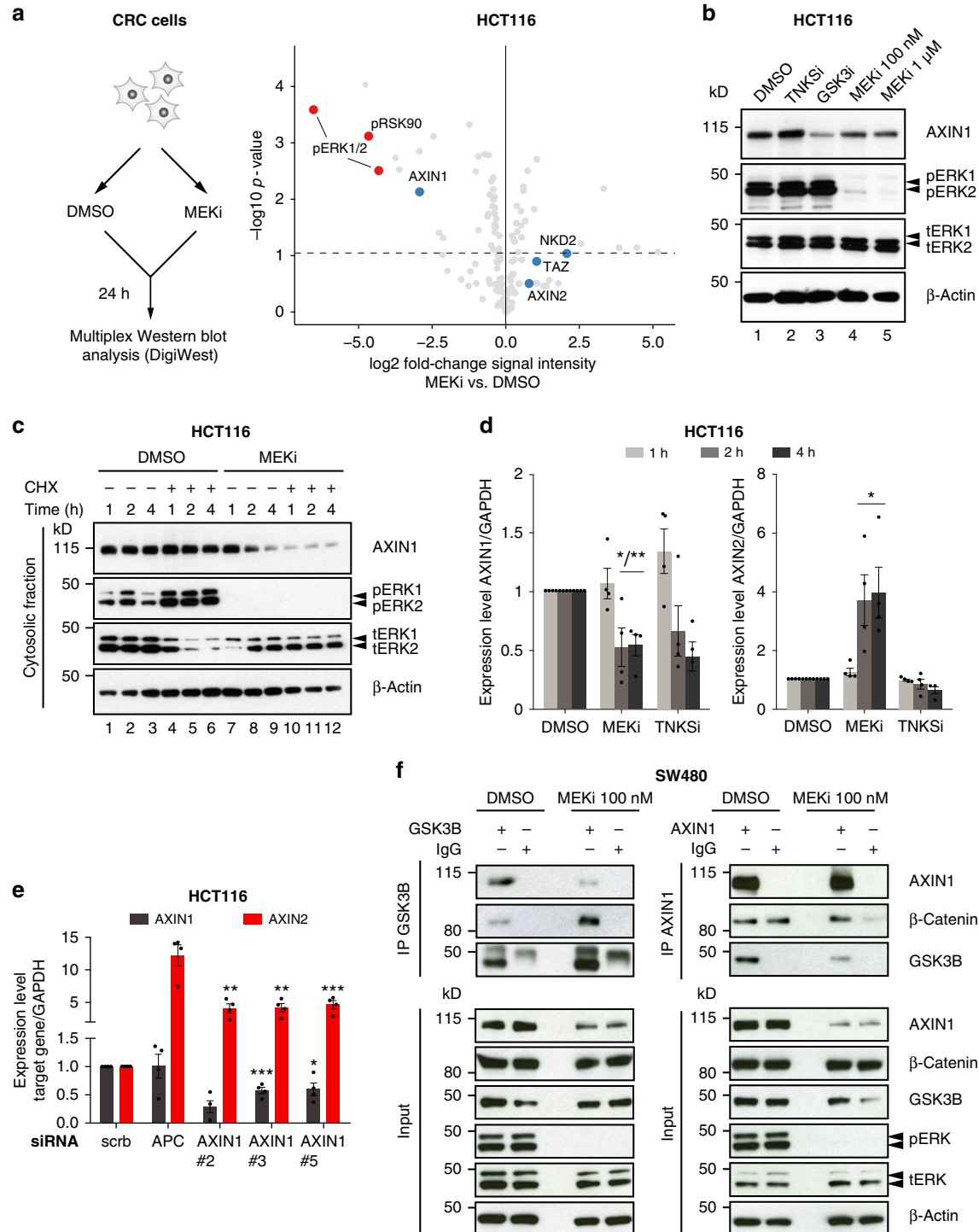

intestine. To address this question, we treated wild-type C57BL/6 mice with intraperitoneal injections of trametinib (2 mg kg$^{-1}$ body weight) or carrier solution (10 mice per treatment condition). After 48 h, mice were sacrificed, and both colon and small intestines were isolated (Fig. 6a). Expression analysis of main Wnt target genes and intestinal stemness markers demonstrated a significant upregulation of *AXIN2*, *CCDN1* and *BIRC5* in the colon (Fig. 6b), and of *LGR5*, *BIRC5*, *CCDN1* (Fig. 6c) and *ASCL2* in the small intestines.

To confirm these results, we used murine intestinal and colon organoid models from mouse strains with different genetic backgrounds. Treatment of intestinal organoids from wild-type

C57BL/6 mice with trametinib resulted in a significant induction of *LGR5* and *ASCL2* expression at 10 and 100 nM (Fig. 6d). To further validate this observation with an alternative model, we used isogenic murine colon organoid lines harbouring an APC truncation and a *KRAS* G12D mutation (APC-KRAS line). To generate this line, colonic crypts were isolated from a C57BL/6 lox-STOP-lox *KRAS* G12D CreERT cross line[29]. Subsequently, the *KRAS* G12D mutation was activated by tamoxifen-induced Cre recombination. An additional APC truncation was introduced by CRISPR/Cas9 genome editing and correctly edited cells were selected using media deprived of Wnt ligands. Compared to the wild-type counterpart, the APC-KRAS line showed an

**Fig. 5** AXIN1 levels are regulated by MEK1/2 inhibitors. **a** A bead-based high-throughput Western Blot (DigiWest) uncovers downregulation of AXIN1 protein levels by MEK inhibition. HCT116 were treated for 24 h with 1 μM of trametinib, lysed and processed for DigiWest. Volcano plot showing signal intensity of selected Wnt and Ras pathway members in trametinib versus DMSO treated cells. Mean values of three independent replicates are shown. Statistical significance was determined by a moderated *t*-test. Protein levels of phospho-ERK and phospho-RSK90 (red dots) are reduced by trametinib, whereas levels of Wnt regulators *AXIN2*, *NKD1* and *TAZ* (blue dots) are increased. In contrast, AXIN1 proteins levels are reduced upon trametinib treatment (see also Supplementary Fig. 5). **b** Western Blot confirms downregulation of AXIN1 protein levels in HCT116. Cells were treated for 24 h with the indicated compounds and then lysed for Western Blot analysis. **c** Trametinib causes a rapid reduction of cytoplasmatic AXIN1 protein level. HCT116 cells were pretreated with 100 μg/ml cycloheximide (CHX) or water for 5 min, followed by addition of DMSO or 100 nM trametinib for 1, 2 or 4 h. AXIN1 levels in the cytosolic fraction were determined by Western Blot. **d** Trametinib rapidly induces *AXIN2* and depletes *AXIN1* expression. HCT116 cells were treated for 1, 2 or 4 h with DMSO, 100 nM trametinib or 10 μM XAV939. Expression of genes were determined by qPCR. **e** siRNA mediated knockdown of *AXIN1* in HCT116 strongly enhances the expression of the Wnt target gene *AXIN2*. HCT116 were transfected with the indicated siRNAs for 48 h and expression of target genes was measured by qPCR. **f** Dissociation of AXIN1 from GSK3B upon trametinib treatment. SW480 were treated for 24 h with 100 nM trametinib and affinity purification of lysates was performed using antibodies against AXIN1 and GSK3B. **b–c**, **f** A representative image of three independent biological replicates is shown. **d–e** Data from four independent experiments are presented as mean ± s.e.m. *$p < 0.05$, **$p < 0.01$, two-sided Student's *t*-test

increased basal expression level of *AXIN2* and the intestinal stemness marker *ASCL2*[30]. While MEK1/2 inhibition did not increase *AXIN2* and *ASCL2* transcript levels in wild-type organoids, treatment of the APC-KRAS organoid line significantly activated the expression of both Wnt target genes (Fig. 6e). This result further supports that APC truncation strongly sensitizes colon cells to the effect of MEK1/2 inhibitors on Wnt signalling. In summary, we show that MEK1/2 inhibition stimulates the expression of Wnt target genes and stemness markers in murine intestine in vivo and in vitro.

**MEK inhibition induces a stem cell program in CRC organoids**. Patient-derived tumour organoids are a potent model of CRC and supposed to reflect cancer biology more adequately than cancer cell lines[31]. Therefore, we tested whether trametinib can transcriptionally induce Wnt signalling and intestinal stemness markers in CRC organoids (Fig. 7a). For this purpose, we obtained endoscopic biopsies of CRC from treatment naive patients and established organoid lines. The mutation status of CRC organoid lines was characterised by amplicon sequencing of frequently mutated genes in CRC (Supplementary Table 2). We selected several cancer organoid lines with divergent genetic backgrounds and treated them with low concentrations of trametinib (1–10 nM) for 72 h. In all cases, an increase of Wnt target genes and intestinal stemness markers, particularly *LGR5* was observed by qPCR (Fig. 7b, Supplementary Fig. 8A). Transcriptome profiling by microarray analysis showed that a set of well-defined Wnt target genes (*AXIN2*, *BIRC5*) and intestinal stemness markers (*LGR5*, *OLFM4*, *ASCL2*, *CD44*, *EPHB2*) were transcriptionally induced upon treatment with trametinib. In contrast, expression of target genes of the RAS pathway (*DUSP5*, *FOS*) and epithelial differentiation markers (*TFF1*, *KRT20*) was repressed (Fig. 7c, Supplementary Fig. 8B). Similar to CRC cell lines, we observed that low concentrations of trametinib were sufficient to reduce phospho-ERK and AXIN1 protein levels (Fig. 7d). Furthermore, we used fluorescence microscopy to show that trametinib could induce a morphological change of cancer organoids resulting in a cystic phenotype, which is characteristic of high Wnt activity[32] (Fig. 7e). We counted the number of cystic organoids on eight images from two independent replicates and observed a significant dose-dependent increase upon trametinib treatment (Fig. 7f).

To obtain a more global view on the transcriptome changes associated with MEK1/2 inhibition, we analysed the microarray data for enrichment of specific gene signatures. A previous study identified expression signatures that are specific for distinct cell types of the intestinal crypt, including intestinal stem cells (LGR5-ISC and EPHB2-ISC), crypt proliferative progenitors (proliferation) and late transient amplifying cells (late TA)[33]. Of

those, the ISC signatures could predict aggressive CRC subtypes with a high risk of disease relapse. Thus, we analysed the transcriptome of a trametinib treated cancer organoid line for enrichment of those signatures by gene set enrichment analysis. We observed a strong enrichment of the EPHB2- and LGR5-ISC signature and a depletion of the proliferation signature, while no change of the TA signature was detected (Fig. 8a–b). This observation indicates that MEK1/2 inhibitor treatment induces an intestinal stem cell program in CRC organoids, which is associated with adverse disease outcome.

Next, we asked whether trametinib induced cellular reprogramming can be reversed by co-inhibition of the Wnt pathway (Fig. 8c). To address this question, we co-treated CRC organoids with trametinib and PRI-724, an inhibitor of β-catenin and CBP interaction that is currently tested in phase II clinical trials for different cancer types[3]. Upon co-treatment, expression of stemness markers and Wnt target genes were repressed while markers of cellular differentiation were re-induced, as confirmed by qPCR and global expression profiling (Fig. 8c–d). Together, these results indicate that additional targeting of the Wnt pathway can prevent trametinib induced cellular reprogramming of CRC.

Finally, we investigated the effect of dual pharmacological inhibition of MEK and Wnt on cancer growth in vitro and in vivo. Co-treatment of HCT116 cells with trametinib and PRI-724 resulted in a dose-dependent, enhanced reduction of cell proliferation compared to treatment with either substance alone (Supplementary Fig. 9A). To test the effect of combinatorial Wnt and MEK inhibition in vivo, we established a PDX model by subcutaneous engraftment of CRC organoids into immunodeficient gamma NOD/SCID mice. We selected a *KRAS* and *APC* mutant organoid line which responded to MEK inhibition with a strong increase in Wnt activity (Fig. 8a–b) and could establish viable tumours with a take rate of 100%. Two weeks after injection of organoids, mice were treated with either trametinib, ICG-001, a combination of both substances or the vehicle solution for a total of 14 days. As shown in Supplementary Fig. 9B, co-treatment with trametinib and ICG-001 resulted in a significant reduction of tumour growth, whereas single agents showed only a minor effect. Interestingly, the majority of mice that were treated with trametinib plus ICG-001 gained body weight, indicating that the combination therapy was well tolerated. Taken together, these results suggest that dual blockage of Wnt and RAS signalling has superior antiproliferative effects in CRC over single inhibition of either pathway.

## Discussion
Treatment of CRC that harbour mutations in the RAS pathway is a major challenge. MEK1/2 inhibitors are a promising option

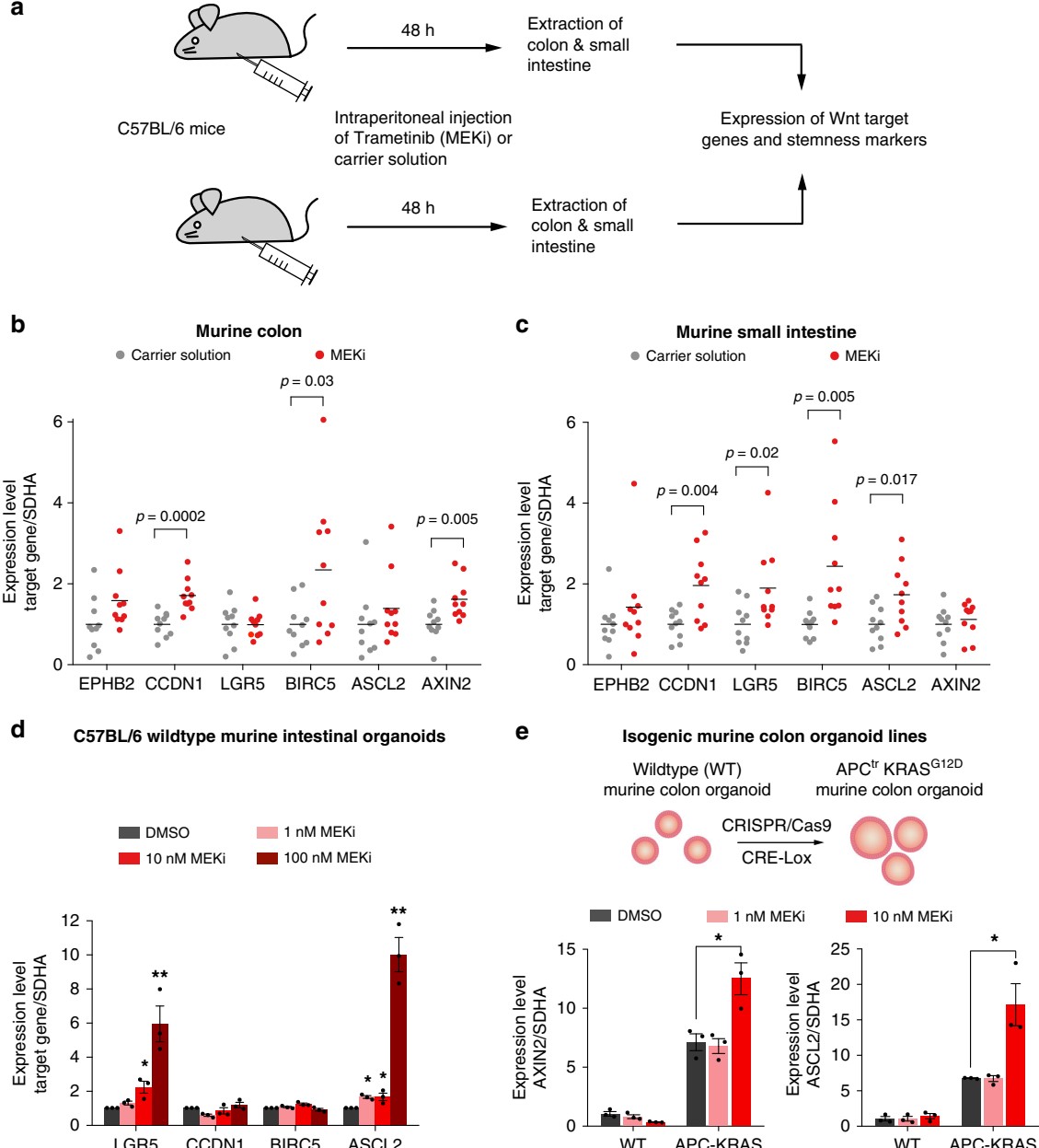

**Fig. 6** MEK inhibition activates Wnt target genes in murine intestine. **a** Schematic overview showing treatment of C57BL/6 mice with the MEK inhibitor trametinib. C57BL/6 mice were treated with either 2 mg kg$^{-1}$ body weight of trametinib or carrier solution. Drugs were administered as intraperitoneal injections. 48 h after injection, animals were sacrificed and the colon and ileum were isolated and processed for further analysis. **b**–**c** Expression of Wnt target genes and stemness markers are increased in the small intestine (**b**) and colon (**c**) of trametinib treated mice. RNA was isolated from tissue of the respective organs and expression levels of target genes were determined by qPCR analysis. Results represent ten animals per treatment group (five male and five female animals). **d** Trametinib increases expression of stemness marker in murine intestinal organoids. Intestinal organoids derived from C57BL/6 mouse were treated for 72 h with different concentrations of trametinib and then lysed for RNA isolation and qPCR analysis. Data from three independent experiments are represented as mean ± s.e.m. **e** Trametinib increases Wnt target gene expression in APC-KRAS mutant colon organoids. Organoids were generated from C57BL/6 lox-STOP-lox *KRAS* G12D CreERT mice. Isogenic (wildtype and APC-KRAS mutant) organoids lines were generated by in vitro Cre-recombination and genome editing of *APC*. Treatment of APC-KRAS, but not wild-type organoids, with trametinib for 72 h caused an increase of *AXIN2* and *ASCL2* expression. Data from three independent experiments are represented as mean ± s.e.m. \*$p < 0.05$, \*\*$p < 0.01$, two-sided Student's *t*-test

to target RAS signalling downstream of oncogenic RAS or RAF[34]. Over the past decade, many MEK1/2 inhibitors have been developed and tested in clinical studies[35]. While they increased progression-free survival of patients with *BRAF* mutant melanoma, they failed to show a similar activity in other solid cancers. In particular, phase II clinical trials demonstrated that treatment with MEK inhibitors could not stop progression of *KRAS* mutated CRC[36]. The biological

processes that underlie this unresponsiveness are not comprehensively understood. Here, we provide several lines of evidence that MEK1/2 inhibitors activate Wnt signalling in CRC. First, by unbiased compound screens, we show that the class of MEK1/2 inhibitors can potently stimulate the Wnt pathway in many RAS mutant CRC lines. Secondly, we used both in vivo and in vitro models to show that this effect is shared across different species and in normal as well as transformed tissue.

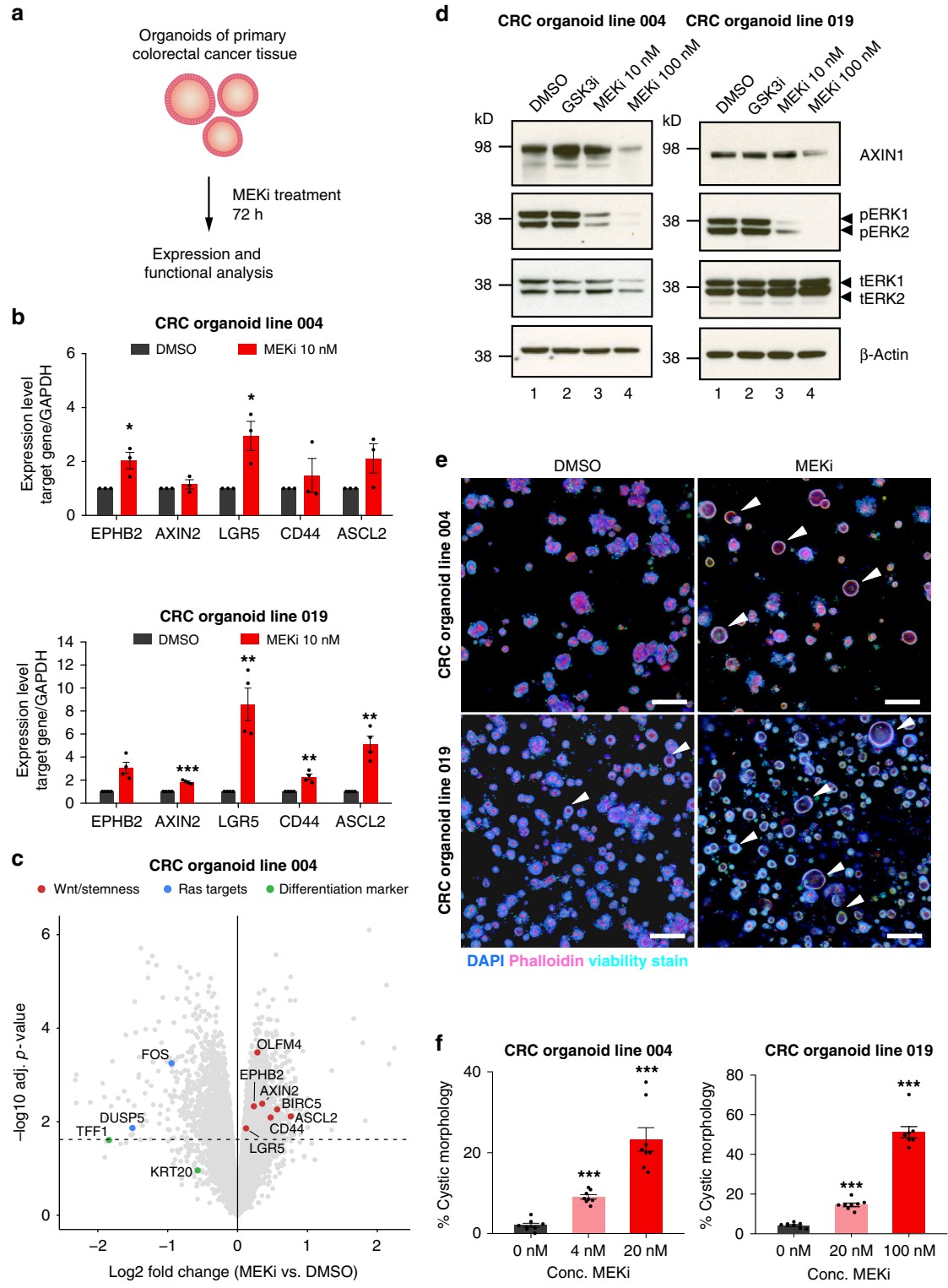

Mechanistically, we provide evidence that MEK inhibition rapidly and persistently reduces AXIN1 protein levels and leads to a dissociation of AXIN1 from GSK3B, thereby stimulating Wnt. Finally, we generated patient-derived CRC organoids to show that LGR5 and intestinal stemness signatures are enriched by MEK inhibitor treatment. Co-targeting of the Wnt and Ras pathway resulted in synergistic reduction of cancer proliferation both in vivo and in vitro. Therefore, we propose that reprogramming of CRC is a yet unknown side effect of RAS pathway inhibition.

Similar to most oncogenic pathways, RAS signalling interacts with itself and other signalling cascades at multiple levels[37]. Upon MEK1/2 inhibition, feedback loops within the RAS pathway are activated, including transcriptional activation of *ERBB3*[38] or heterodimerization of BRAF and RAF1[39]. Moreover, other signalling pathways were found to compensate for reduced RAS signalling, including Hippo[40] or JAK/STAT[41]. Combining MEK with EGFR/RAF inhibitors or small molecules targeting one of the salvage pathways synergistically reduced viability in vitro[38,39]. Based on these findings, several clinical trials were initiated to

**Fig. 7** MEK inhibition induces Wnt signalling and stemness in colorectal cancer organoids. **a** Overview of experimental setup. A panel of patient-derived colorectal cancer organoids was grown in ENA medium, treated for 72 h with different concentrations of trametinib and then analysed by functional and expression assays. **b**–**c** MEK inhibition induces Wnt target genes and intestinal stemness markers. **b** Differential expression of selected Wnt target genes and stemness markers are induced by MEK inhibition, as determined by qPCR. **c** Expression analysis with microarray of colon cancer organoids treated with trametinib shows increased expression of many of Wnt and stemness associated genes, and a reduction of RAS target genes and differentiation markers (**b**). **b**–**c** Data from three independent experiments are presented as mean ± s.e.m. **d** MEK inhibition leads to reduction of AXIN1 levels in colorectal cancer organoids. Colorectal cancer organoids from two patients were treated with the trametinib for 72 h and levels of indicated proteins were analysed by Western Blot. A representative Western Blot of three independent replicates is shown. The effect of MEK inhibition on gene expression in additional CRC organoid lines is shown in Supplementary Fig. 8. **e**–**f** Treatment with trametinib induces a Wnt phenotype in human colorectal cancer organoids. Organoids were treated for 48 h with indicated concentration of drugs, fixed with PFA and stained with DAPI (blue), phalloidin (red) and Image-IT DEAD Green Viability Stain (green). White arrows indicate organoids with a cystic morphology, which is associated with high Wnt activity. Scale bar: 200 μm. **e** Quantification of the percentage of cystic organoids shows a dose dependent increase upon treatment with trametinib (**f**). Results of eight images per concentration of two independent replicates are shown as mean ± s.e.m. *$p < 0.05$, **$p < 0.01$, ***$p < 0.001$, two-sided Student's $t$-test

assess the effect of combinatorial treatments[42,43]. However, tumour remissions were only achieved in a minor fraction of patients[43], indicating that more complex cellular processes are activated upon interference with MEK.

In our study, we demonstrate that activation of Wnt signalling is a specific cellular feedback mechanism that is triggered by inhibition of MEK in RAS/RAF mutant CRC. Our findings are surprising, as most crosstalks between the RAS and Wnt pathway have been described as mutually activating[44–47]. However, studies in *BRAF* mutant CRC and melanoma indicate that Wnt signalling can be activated upon targeting of BRAF[48,49]. Here, we show that the feedback loop relies mechanistically on AXIN1 as an intersection point between both pathways. AXIN1 is the central scaffold of the destruction complex and a concentration-limiting protein for the assembly of the complex[22]. Cellular levels of AXIN1 are regulated by posttranslational modifications, including degradation by activated LRP5/6[50] and tankyrases[23], and stabilization by SUMOylation[51]. Furthermore, the interaction of AXIN with GSK3B is essential for the integrity of the destruction complex and a dissociation of AXIN1 from GSK3B was observed upon Wnt stimulation[28]. Here, we show that MEK inhibition regulates AXIN1 by rapid transcriptional repression and propose that downregulation of the transcription factor *EGR1* could be a potential link to RAS activity. We also observed that depletion of AXIN1 protein levels by MEK inhibition is enhanced if global translation is blocked, indicating that AXIN1 undergoes a dynamic turnover which requires permanent de novo synthesis to maintain stable protein levels. Furthermore, we observed a dissociation of AXIN1 from GSK3B as a second potential mode of Wnt activation. Whether these mechanisms work independently or concurrently to stimulate Wnt signalling is yet to be determined. Surprisingly, we also observed that APC truncations synergize with MEK and ERK inhibition to activate Wnt, while they completely abolish the stimulating effect of GSK3B inhibitors. This finding indicates that GSK3B and MEK/ERK act on Wnt signalling by distinct mechanisms, which can be independently targeted by small molecules. Although *APC* mutations remain the most important driver of Wnt signalling in CRC[52], many Wnt components can modulate the signal output in spite of APC truncations. For instance, interference with Wnt secretion by depletion of Wls/Evi[53] or epigenetic silencing of the extracellular Wnt antagonist SFRP[54] can affect Wnt activity. Furthermore, targeting components of the destruction complex such as CK1α[55] or AXIN[23] can also modulate Wnt signalling in *APC* mutant CRC. Finally, secreted cues from the tumour microenvironment such as hepatocyte growth factor[6] or exosomes[56,57] can locally affect Wnt activity and stemness in CRC. It is therefore promising to further dissect the intrinsic and extrinsic signals that modify Wnt signalling in CRC with *APC* mutations, with the ultimate goal to identify druggable targets that reduce aberrant Wnt signalling.

Our study shows that a minimal inhibition of RAS signalling is sufficient to activate an intestinal stem cell program in CRC that is marked by increased expression of well-known stem cell markers such as *ASCL2*[30], *EPHB2*[33] or *LGR5*[58]. The important role of *LGR5* for CRC stem cells has emerged recently. LGR5+ CRC cells were shown to be tumour initiating cells in murine CRC and required for the development and maintenance of liver metastasis[9]. In human CRC, LGR5+ cells have self-renewal and differentiation capacity and fuel tumour growth as cancer stem cells. However, upon ablation of LGR5+ cells, differentiated KRT20+ cancer cells can revert to LGR5+ cells and drive cancer regrowth[59]. This observation supports a revised model of cancer stemness which proposes that cancer cells can dynamically shift between a differentiated and a stemness state as a response to environmental perturbations[7]. Our study provides further experimental evidence supporting this model of cancer stem cell plasticity. We demonstrate that inhibition of MEK1/2 activates Wnt signalling and elicits transcriptome changes that suggest a transition from differentiation to stemness in CRC organoids (see model in Fig. 8e). However, we also observed that MEK inhibitors reduce proliferation rates, as it was shown in many models of CRC[60]. This diminished cell growth—together with high Wnt and stemness activity - results in a phenotype that is reminiscent of quiescent cancer stem cells. Quiescent or slow-cycling tumour stem cells have been recently identified in many cancer types and emerged as an important mediator of therapy resistance[7]. These stem cells are able to survive antineoplastic treatment and fuel cancer regrowth upon release from therapy[61,62]. Based on our experimental data, we propose that MEK1/2 inhibition induces cellular plasticity resulting in enrichment of quiescent cancer stem cells. This hypothesis is supported by transcriptome changes activated by MEK inhibition, which are enriched for intestinal stem cell gene signatures that are predictive of disease relapse[33]. Our data indicates that activation of Wnt signalling upon targeting of MEK is a salvage mechanism that facilitates survival of CRC, which could explain the failure of MEK inhibitors as monotherapy in this cancer entity[63,64]. This hypothesis is supported by our study as we show that combined pharmacological targeting of Wnt and MEK is more efficient in reducing tumour growth than single agent treatment, in both CRC cells and a patient-derived cancer xenograft model. In line with our observations, recent studies demonstrated that EGFR inhibition can induce quiescence of LGR5+ murine intestinal stem cells[8] and that treatment with cetuximab and genetic ablation of LGR5 synergistically enhanced CRC eradication[59]. Therefore, we propose that a combination therapy of MEK and Wnt inhibitors, such as PRI-724, could be a promising approach for the treatment of CRC.

In summary, our work demonstrates that clinically used MEK inhibitors induce Wnt signalling and plasticity of cancer stem cell

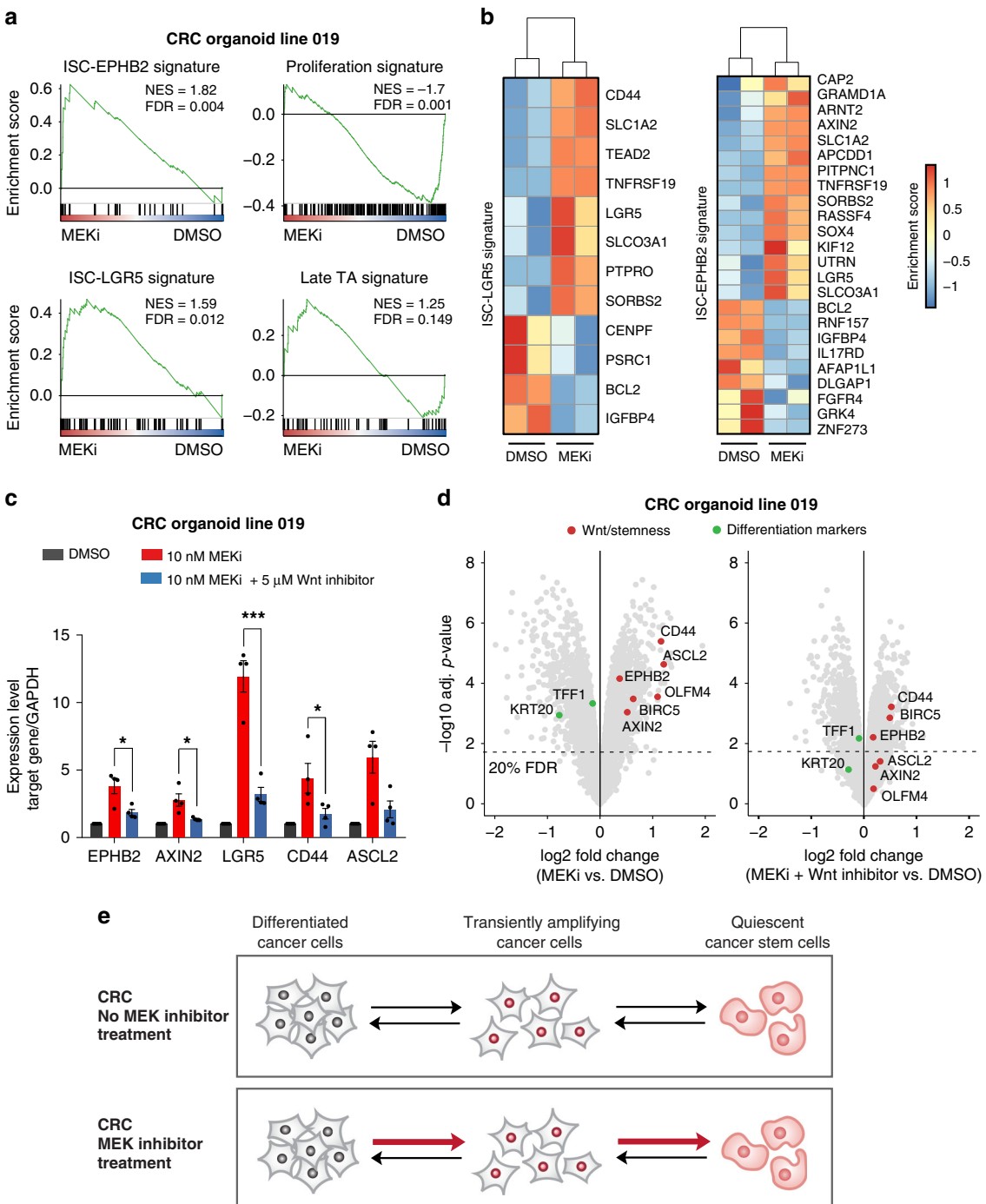

**Fig. 8** MEK inhibition induced stemness signatures are associated with CRC relapse. **a–b** Trametinib treatment induces expression signatures in human colon cancer organoids associated with intestinal stemness. Colorectal cancer organoids were treated for 72 h with trametinib and the global expression pattern was determined by microarray. Transcriptome data of treated human colon cancer organoids were analysed for enrichment of three previously described gene signatures by gene set enrichment analysis (GSEA). **b** Heatmap depicting the fold-change of individual genes within the humanised ISC-LGR5 and the ISC-EPHB2 gene signature. Data from two biological replicates are shown. **c–d** Co-treatment of colon cancer organoids with Wnt inhibitor PRI-724 abolishes the trametinib induced Wnt and stemness activation. **c** Human colorectal cancer organoids were treated with 10 nM of trametinib and 5 μM of the Wnt inhibitor PRI-724. Expression of selected Wnt target and stemness genes was analysed by qPCR. The effect of co-treatment on an additional CRC organoid line is shown in Supplementary Fig. 8. Data from four independent experiments are represented as mean ± s.e.m. *p < 0.05, ***p < 0.001, two-sided Student's t-test. **d** Human colorectal cancer organoids were treated with 10 nM of trametinib and 10 μM of the Wnt inhibitor PRI-724. Global expression was measured by microarray. Data from two independent experiments are shown. **e** Model of MEK inhibition induced Wnt-activation. Inhibition of the MEK1/2 leads to activation of Wnt signalling and intestinal stemness signatures. This change of signalling networks results in the induction of cellular plasticity and reprogramming of differentiated cancer cells into cancer stem cells

in CRC. This crosstalk provides a potential target for rational combination therapy.

## Methods

**Ethics approval.** Collection of tissue biopsies, generation of cancer organoids and experiments with organoids were approved by the Medical Ethics Committee II of the Medical Faculty Mannheim, Heidelberg University (Reference no. 2014–633N-MA and 2016–607N-MA). All patients gave written informed consent before performing any experimental procedures.

**Small molecule inhibitors.** Trametinib, ICG-001, PRI-724, dabrafenib, GDC-0994, PD318088, selumetinib were purchased from Selleckchem. Erlotinib, SCH772984 and RAF265 were purchased from Cayman Chemical. Gefitinib was purchased from BioVision. XAV939 and CHIR99021 were purchased from Merck Millipore.

**Cell lines and culture.** HCT116, HT29, SW480, RKO, DLD1, HEK293T cells were derived from ATCC and SW403 from DSMZ. HCT116 and HT29 cells were cultured in McCoy's medium (Life Technologies). HEK293T, DLD1 and RKO cells were cultured in DMEM medium (Life Technologies). SW480 and SW403 cells were cultured in RPMI medium (Life Technologies). All media were supplemented with 10% foetal calf serum (PAA). For compound screens, 1% v/v penicillin/streptomycin (Life Technologies) was added to the media. Cell lines were obtained from ATCC and authentication of genotypes was performed by SNP profiling (Multiplexion, Heidelberg). Absence of mycoplasma infection was confirmed by regular testing.

**Generation of Wnt reporter cell lines for compound screens.** To generate CRC cell lines with stable TCF/Wnt-reporter expression, we used the lentiviral plasmids 7TFP[18] and TCF4/Wnt[53]. Viral particles were produced by transfecting HEK293T with the lentiviral packaging plasmids pMD2.G (#12259, Addgene) and psPAX2 (#12260, Addgene) together with the vector of interest. Supernatants were collected after 48–72 h and sterile-filtered (45 μm). Target cells were infected in T75 flasks by addition of 1–2 ml viral supernatant and Polybrene Transfection reagent (1:1000) (Merck Millipore). After 72 h, transduced cells were selected by addition of 1–2 μg ml$^{-1}$ puromycin. Aliquots of selected cells were stored at −80 °C and same batches of cells were used for compound screens.

**Small molecule screen.** The Biofocus (BioFocus) and Spectrum (Microsource Discovery Systems) compound library were prediluted in RPMI to obtain a concentration of 50 μM. Additional compounds with known Wnt modulating activity and DMSO were added as controls. For the focused screen, the kinase inhibitor library (Selleckchem) was diluted with RPMI to a final concentration of 100 μM. A list of all compounds is provided in Supplementary Data 1–2. Cells were plated on white 384 well LIA plates (Greiner) with the following concentrations: 1000 cells of HCT116-TCF4, 1500 cells of DLD-7TFP and DLD-TCF4, 2800 cells of SW480–7TFP in 45 μl medium per well. Cell seeding was performed using a Multidrop Combi (Thermo Fisher). Two sets of plates were generated, one for luciferase read-out and the other for measurement of cell viability using the CellTiterGlo assay (Promega). Twenty-four hours post seeding, 5 μl of compound solution was added to the plates using a Biomek FX robot (Beckman Coulter) with a 384 well tip head to obtain a final concentration of 5 μM (Biofocus and Spectrum library) and 10 μM (kinase inhibitor library). Cells were then cultured for another 24 h prior to analysis. The luciferase read-out and CellTiterGlo assay were performed separately on an automated Tecan freedom Evo workstation (Tecan). Media were removed from the plates using a Biotek Infinite EL x405 CW washer. Lysis buffer/firefly-luciferase or CellTiterGlo solution (diluted 1:5 in PBS) was added to each well using a Multidrop Combi (Thermo Fisher Scientific). Luminescence signals were determined using a Tecan F500 reader (Tecan). Raw photon count files were processed with webcellHTS2[65] with the following predefined settings: normalization method—median, log-transformation of raw values—no, normalization scaling method—additive, adjustment of variance—by plate, summary of replicates—mean. In brief, raw firefly-luminescence and CellTiterGlo signals of each well were normalised to the median signal of all samples on the respective plate. Two independent replicates were performed for each compound screen and normalised data for CellTiterGlo and firefly luminescence of each replicate is presented in Supplementary Data 1, 2. The Wnt/luciferase activity of each replicate was calculated by dividing the normalised firefly-luminescence by the normalised CellTiterGlo signal. The mean Wnt/luciferase activity of both biological replicates are presented in Fig. 1.

**Digital Western Blot (DigiWest).** DigiWest was conducted as published in Treindl et al.[21]. Gel electrophoresis and blotting onto PVDF membranes was performed with the NuPAGE system (Life Technologies). Blots were washed in PBST, proteins were biotinylated on the membranes using NHS-PEG12-Biotin (50 μM) in PBST for 1 h, washed in PBST and dried. Each sample lane was cut into 96 molecular weight fractions of 0.5 mm heights and proteins were eluted in 96 well plates using 10 μl elution buffer per well (8 M urea, 1% Triton-X100 in 100 mM Tris-HCl pH 9.5). The eluted proteins of each molecular weight fraction were

bound onto distinct colour-coded, neutravidin coated Luminex beads (MagPlex, Luminex) and subsequently pooled. Fifteen micrograms of protein were used for the DigiWest procedure and bead sets for 200 antibody incubations were generated.

Aliquots of the DigiWest bead-mixes (about 1/200th per well) were added to 96 well plates containing 50 μl assay buffer (Blocking Reagent for ELISA (Roche) supplemented with 0.2% milk powder, 0.05% Tween-20 and 0.02% sodium azide) per well. After discarding the assay buffer 30 μl of diluted antibody was added per well. After overnight incubation at 15 °C on a shaker, the bead-mixes were washed twice with PBST and PE-labelled (Phycoerythrin) secondary antibodies (Dianova) were added and incubated for 1 h at 23 °C. Beads were washed twice prior to readout on a Luminex FlexMAP 3D. Secondary antibodies were either diluted in assay buffer or in a polymer buffer (Blocking Reagent for ELISA (Roche) supplemented with 4% PVP 360.000, 1% milk powder, 0.05% Tween-20 and 0.02% sodium azide). All antibodies used are shown in Supplementary Table 1.

For quantification of the antibody specific signals the analysis tool described in Treindl et al.[21] was used that identifies peaks of appropriate molecular weight and calculates the peak area. Results are presented as median ± SE, unless otherwise specified. For comparative analysis, the obtained protein expression values were normalised to the total protein amount loaded to one lane.

**Quantitative PCR.** Total RNA was isolated from cells using the RNeasy Mini kit (Qiagen). cDNA was synthesised with the RevertAid H Minus First Strand cDNA Synthesis Kit (Thermo Fischer Scientific) using 1 μg of total RNA as a template. Quantitative PCR was performed in a 384 well plate on a LightCycler 480 device (Roche) using the Universal probe library system (Roche). GAPDH (human samples) and SDHA (murine samples) were used as reference genes for relative quantification. Primers used for qPCR are listed in Supplementary Table 3.

**Immunoblot analysis.** Protein extraction was performed with Triton-containing lysis buffer (20 mM Tris-HCl pH 7.4, 130 mM NaCl, 2 mM EDTA, 1% Triton X-100 0.1%) supplemented with protease inhibitor tablets (Roche) and phosphatase inhibitor cocktails 1 and 2 (Sigma-Aldrich). Protein concentration was determined by BCA protein assay (Thermo Fisher Scientific). Ten to thirty micrograms of lysates were loaded on 4–12% NuPAGE Bis-TRIS gels (Life Technologies) and transferred to Immobilon PVDF membranes (Millipore). If needed, membranes were additionally treated with Pierce Western Blot Signal Enhancer Kit (Fisher Scientific Scientific). A list of all antibodies used in this work and dilutions can be found in Supplementary Table 4. Contrast of Western Blot images was adjusted using Adobe Photoshop and uncropped, unprocessed scans are found in the Source Data file.

**Wnt reporter assay.** To quantify Wnt signalling activity in RKO and RKO-APC cells, dual expression of reporter plasmids was used. For this approach, cells were seeded in a 384 well white, flat-bottom polystyrene plate (Greiner) at a density of 7500 cells per well. After 24 h, cells were transfected with 20 ng of actin-renilla and 40 ng of TCF4/Wnt firefly luciferase reporter constructs. Twenty-four hours after reporter plasmids transfection, cells were treated with CHIR99021 (Merck Millipore), XAV939 (Sigma Aldrich) or trametinib (Selleckchem). Reporter levels were measured 48 h after transfection using a Mitras LB940 plate reader (Berthold Technologies) and the TCF4/Wnt-luciferase signal was normalised to actin-Renilla luciferase reporter signal.

**Cycloheximide chase assay.** $2 \times 10^5$ HCT116 cells per well were seeded in a six-well plate. Twenty-four hours post seeding, cell culture medium was replaced with McCoy's medium containing either 100 μg/ml cycloheximide (Sigma) or water. After 5 min, cells were treated with DMSO or 100 nM trametinib. Cell pellets were collected 1, 2 and 4 h after addition of trametinib and resuspended in saponin containing lysis buffer (20 mM Tris-HCl pH 7.4, 130 mM NaCl, 2 mM EDTA, 10 mM β-mercaptoethanol, 0.05% saponin) supplemented with protease inhibitors tablets (Roche) and phosphatase inhibitor cocktails (Sigma-Aldrich) for cytoplasmic protein extraction. Protein samples were analysed by Western Blot.

**RNA interference (RNAi).** HCT116 cells were seeded at a density of $2 \times 10^5$ cells per well on six-well plates (Greiner). Twenty-four hours after cell seeding, cells were transfected with siGenome Upgrade siRNAs (Dharmacon) and DharmaFECT1 (Dharmacon) or RNAiMAX (Thermo Fisher Scientific) with a final concentration of 10 nM siRNA per well. For subsequent expression analysis, cells were harvested 72 h post transfection. If additional drug treatment was performed, the medium containing siRNAs was removed 48 h after transfection and replaced by fresh McCoy's medium supplemented with the respective drugs. Drug treatment was applied for 24 h. Sequences of siRNAs used are listed in Supplementary Table 5.

**Plasmid transfection.** $2 \times 10^5$ HCT116 cells per well were seeded on six-well plates. Twenty-four hours post seeding, cell transfection was performed using TransIT-LT1 transfection reagent (Mirus) according to the manufacturer's protocol. For overexpression experiments, 500 ng of control plasmid mCherry2-N1

(Addgene, #54517) or 500 ng of cDNA of human EGR1 (Biocat, #SC128132-OR) plasmid were used. Forty-eight hours after transfection, cells were treated for 4 h with either 100 nM trametinib or DMSO. Cell pellets were then harvested to extract total cellular RNA.

**Genome editing of APC truncation.** The sgRNA targeting the APC gene was designed using the E-CRISP sgRNA design tool[66]. The selected sgRNA sequence was (bold letters indicate PAM sequence):

sgAPC 5′-TCTGCTGGATTTGGTTCTA**GGG**−3′.

Pairs of oligonucleotides encoding the sgRNA were synthesised by Eurofins Inc. Oligonucleotides were phosphorylated, annealed and cloned into a Bbs1 digested px459 plasmid (#62988, Addgene) using Quick Ligase (NEB). To generate an APC truncation, RKO cells were transiently transfected with 2 µg of px459 with sgAPC. After 48 h, cells were selected with 1 µg/ml of puromycin for 48 to 72 h. Single clones were generated by serial dilutions in 96 well plate. After 10–15 days, colonies derived from single clones were expanded for further analysis. Targeted deep sequencing of PCR amplified APC genomic locus was performed to assess indel formation introduced by the sgRNA. Genomic DNA of APC mutant single cell clones was isolated using DNeasy Blood and Tissue Kit (Qiagen). Primer pairs were designed 100 to 150 bp up and downstream of the sgRNA targeting site and adapters were added during the second step of the nested PCR. The PCR primers for the first PCR step were 5′- TCCCTACACGACGctcttccgatctTCAGACGACAC AGGAAGC-3′ and 5′- AGTTCAGACGTGTGctcttccgatctACATAGTGTTCAGGT GGACT-3′. The resulting PCR products were purified with the PCR Clean-up Kit (Machery-Nagel) and amplified with a second PCR step to introduce unique indexes. The second PCR reaction was purified using Agencourt Ampure XP Beads (Beckman Coulter) and samples were sequenced on a MiSeq (Illumina) by the Genomics and Proteomics Core Facility of the DKFZ. The multiple sequence alignment tool ClustalOmega[67] was used to analyze indel formation.

**CRISPR interference experiments.** Three sgRNA sequences targeting the transcriptional start site of AXIN1 were selected from a previously published CRISPRi library[68] and were as follows (bold letters indicate PAM sequence):

sgAXIN1 #1: 5′-GCTGGCTCGGGAGGCGA**TCG**−3′
sgAXIN1 #2: 5′-GCTCCGGCCGGCTCTGG**CGG**−3′
sgAXIN1 #3: 5′-GCGACTCCGGCCGGCTC**TGG**−3′

Pairs of oligonucleotides encoding the sgRNAs were synthesised by Eurofins Inc., annealed and cloned into a Bbs1 digested pMarie-V2 plasmid. The pMarie-V2 plasmid is based on the published vector pMarie[69], but encodes an improved sgRNA scaffold[70]. SW480 cells were infected with lentiviral particles encoding the Lenti dCas-KRAB-blast plasmid (gift from S. Diederichs[71]) and pMarie-V2 plasmid with sgRNAs targeting AXIN1. After selection with 2 µg/ml puromycin and 10 µg/ml blasticidin, stable AXIN1 knockdown cells were used for Western Blot analysis or quantitative PCR.

**Affinity purification and mass spectrometry.** For affinity purification, $9 \times 10^6$ cells were seeded in four 15 cm dishes. After 24 h, two 15 cm dishes were treated with 100 nM trametinib and two with DMSO. Twenty-four hours post treatment, cells were washed twice with cold PBS and incubated on ice in lysis buffer (20 mM Tris-HCl, pH 7,4, 130 mM NaCl, 2 mM EDTA, 1% Triton X-100, proteinase and phosphatase inhibitors cocktail) for 10 min, scraped off the plates and placed on ice. Lysates were sheared 10× using a 27-gauge needle and incubated for 10 min on ice. Then, the lysates were centrifuged at $16,000 \times g$ and 4 °C for 30 min, after which the supernatants were recovered and stored on ice. Meanwhile, Dynabeads Protein G magnetic beads (Invitrogen) were washes twice with PBS and protein lysis buffer. Per experimental condition, 50 µl of washed Dynabeads and 1 µg of AXIN1 antibody (Cell Signalling) or rabbit IgG control antibody (Santa Cruz) were added to the stored supernatant and incubated for 12 h at 4 °C on a rotating wheel. Beads were washed repeatedly with lysis buffer and proteins were eluted by addition of 50 µl of 2× Laemmli buffer.

Eluted peptides were subsequently loaded on a SDS gel. After a short run (0.5 cm), the total sample was cut out unfractionated and used for trypsin digestion according to modified protocol by Shevchenko et al.[72]. To enable relative quantification of samples, peptides have been chemically modified with stable isotope dimethyl labelling according to Boersema et al.[73]. A heavy and its corresponding light sample were pooled before LC-MS analysis and loaded on a cartridge trap column, packed with Acclaim PepMap300 C18, 5 µm, 300 Å wide pore (Thermo Scientific). Peptides were subsequently eluted and separated in a 2 h gradient ranging from 3 to 40% ACN on a nanoEase MZ Peptide analytical column (300 Å, 1.7 µm, 75 µm × 200 mm, Waters) and directly analysed by a Q-Exactive-HF-X mass spectrometer (Thermo Scientific). A data dependent scan strategy was applied, in which one full scan at 60 k resolution was followed up by up to 10 MSMS scans at 15 k resolution of the most abundant peptides. To minimize oversampling a dynamic exclusion time of 60 s after peptide fragmentation was activated.

MS raw data was further processed by the MaxQuant software package, version 1.6.0.16[74] mostly using default settings. The cut-off for identification has been based on a false discovery rate of <1%, both on peptide and protein level. "Match

Between Runs" option was enabled, allowing to transfer identifications if the same intact peptide mass was found at the same retention time in different raw files.

For protein quantification at least two peptides were required to be quantified in between samples. Furthermore "Requantify" option was enabled to force quantification of peptides with extremely high abundance differences between samples.

**Culture of murine intestinal and colon organoids.** Murine organoids were isolated and cultured as described by Sato et al.[31]. In brief, murine intestinal and colon crypts were isolated with EDTA treatment of washed tissue samples. Developing organoids were cultured in droplets of Matrigel (Fisher Scientific) and the medium exchanged every 48–72 h. The medium for colon organoids contained 30% basal medium (advanced DMEM/F12 (Life Technologies) with 1% v/v penicillin/streptomycin solution (Life Technologies), 1% v/v HEPES buffer (Life Technologies) and 1% v/v Glutamax (Life Technologies), 50% Wnt3A conditioned medium, 20% R-spondin conditioned medium, 100 ng/ml recombinant noggin (Peprotech), 1x B27 (Life Technologies), 1.25 mM n-acetyl-cysteine (Sigma-Aldrich), 10 mM nicotinamide (Sigma-Aldrich), 50 ng/ml EGF (Peprotech), 500 nM A83–01 (Tocris), 3 µM SB202190 (Biomol), 10 µM Y-27632 dihydrochloride (Sigma) and 100 µg/ml primocin (Invivogen). For APC-KRAS colon organoid lines, the Wnt and R-spondin conditioned medium was replaced by basal medium. For the intestinal organoid medium, the Wnt conditioned medium was replaced by basal medium and A83–01 and SB202190 were removed.

**Culture of patient-derived cancer organoids.** Patient-derived CRC organoids were cultured in droplets of Matrigel and the medium was exchanged every 48–72 h. The culture medium (ENA) contained basal medium (advanced DMEM/F12 (Life Technologies) with 1% v/v penicillin/streptomycin solution (Life Technologies), 1% v/v HEPES buffer (Life Technologies) and 1% v/v Glutamax (Life Technologies), 100 ng/ml recombinant noggin (Peprotech), 1× B27 (Life Technologies), 1.25 mM n-acetyl-cysteine (Sigma-Aldrich), 10 mM nicotinamide (Sigma-Aldrich), 50 ng/ml EGF (Peprotech), 500 nM A83–01 (Tocris), 10 nM gastrin (Peprotech), 10 nM prostaglandin E2 (Santa Cruz Biotechnology), 10 µM Y-27632 dihydrochloride (Sigma) and 100 mg ml$^{-1}$ Primocin (Invivogen).

**Imaging of patient-derived organoids.** Patient-derived cancer organoids were seeded on 384 wells plates coated with BME (Trevigen). 72 h after seeding, organoids were treated with different concentrations of trametinib or DMSO for 96 h. Organoids were then fixed with 4% PFA (Sigma), stained with Phalloidin-TRITC (Sigma), DAPI (Sigma) and Image-IT DEAD Green Viability Stain (Thermo Fisher). Images were obtained on an InCell 6000 automated line-scanning confocal fluorescence microscope (GE Healthcare). Contrast of images was adjusted using ImageJ.

**Mouse experiments.** C57BL/6 mice (6 months of age, average body weight 25 g, five male and five female animals) received i.p. injections of the MEK inhibitor trametinib or carrier solution (DMSO in PBS), respectively. Trametinib dissolved in sterile DMSO and diluted in sterile PBS was administered as a single dose of 2 mg kg$^{-1}$ body weight. After 48 h, mice were anaesthetised, sacrificed and organs of interest were isolated. The organs were snap-frozen in liquid nitrogen for qPCR. The animal studies were performed in agreement with the ethical guidelines of Heidelberg University for animal testing and research. The experiments are approved by the local Governmental Committee for Animal Experimentation (RP Karlsruhe, Germany license G-176/12, G-146/15, G-188/18 to E.B. / Medical Faculty Mannheim, University of Heidelberg).

**Patient-derived CRC xenograft experiments.** All animal experiments were in accordance with the approved guidelines of the responsible national authority, the local Governmental Committee for Animal Experimentation (RP Karlsruhe, Germany, license G-148/18 to T.Z. and K.M-D. / German Cancer Research Center). Mice were maintained at a 12 h light-dark cycle with unrestricted Kliba diet 3307 and water. Under isoflurane inhalation anaesthesia (1–1.5% in air, 0.5 l min$^{-1}$), $2 \times 10^6$ CRC organoid cells were suspended in 75 µl of Matrigel (Fisher Scientific), thus resulting in a final volume of 100 µl, and injected subcutaneously into the right flank of 6–7-week-old female NOD SCID gamma (NSG) mice recruited from the Center for Preclinical Research, DKFZ, Heidelberg. Mice were randomised 14 days after transplantation when tumours reached a volume of mean of 0.09 + 10% cm³. Immediately thereafter, the treatment of $n = 7$ mice/group was started by oral gavage with either 10 µl g$^{-1}$ bodyweight of a 1:1 (v/v) mixture of vehicle of trametinib (final concentration: 0,5% methocel [Sigma], 0.2% Tween 80 in Aqua Braun, pH 8): vehicle of ICG-001 (final concentration 1% carboxymethylcellulose [Sigma] in Aqua Braun), 1 mg kg$^{-1}$ trametinib, 200 mg kg$^{-1}$ ICG-001 or the combination of trametinib and ICG-001 (1 mg kg$^{-1}$ and 200 mg kg$^{-1}$) once daily for 5 days and 2 days off. In total, 14 applications were done. Tumour volume was measured with a caliper two to three times a week and calculated according to the formula: $V =$ (length (mm) × width (mm)²)/2. Necropsies were taken 24 h after the last treatment or when one tumour diameter reached 1.5 cm. Tumours were snap-frozen in liquid nitrogen and stored at −80 °C. No mice were lost due to adverse inhibitor effects. Mice did not lose weight during treatment.

**Expression profiling and gene set enrichment analysis**. Patient-derived organoids embedded in Matrigel (Corning) were lysed using Buffer RLT supplemented with 1% v/v β-mercaptoethanol and the RNA was isolated using the RNeasy Mini Kit (Qiagen). Expression profiling of RNA using the HumanHT-12 v4 Expression BeadChip Kits (Illumina) or the GeneChip Human Genome U133 Plus 2.0 Array (Thermo Fisher) was performed by the Genomics and Proteomics Core Facility of the German Cancer Research Center. All experiments were performed in two to three biological replicates.

Raw intensity values measured with the HumanHT-12 v4 Expression BeadChip were normalised using the 'lumi' R/Bioconductor package[75]. Specifically, background correction was performed using the robust multi-array analysis (RMA) method[76] followed by variance stabilization with the 'vst' method as implemented in the 'lumiT' function. Quantile normalization was performed to adjust the distributions of expression values between microarrays. Raw data measured with the Affymetrix U133 Array were processed using the 'affy' R/Bioconductor package[77]. Again, background correction was performed by RMA[76], and quantile normalization was applied to adjust the distributions of expression levels between arrays. Additionally, PM intensities were corrected for non-specific binding and expression values were summarised using the 'medianpolish' method. Differential gene expression between treatment conditions was analysed at the probe level using a moderated $t$-test as implemented in the 'limma' R/Bioconductor package[77]. To generate the volcano plots, the probes with the highest absolute difference in expression levels between treatment conditions were selected to represent each gene. For the enrichment analysis of intestinal stem cell signatures 'Humanised Crypt Cell Signatures' were downloaded from Merlos-Suarez et al.[33]. Gene set enrichment analyses were performed using the 'fgsea' R/Bioconductor package[78].

**Long-term viability assay**. HCT116 cells were seeded at a concentration of 30,000 cells per well in 12-well plates. Twenty-four hours post seeding, cells were treated with either DMSO or different concentrations of PRI-724 or trametinib or combinations of both compounds for 6 days. Media containing the drugs was renewed after 3 days of treatment. After 6 days, cells were further cultivated in media without drugs. Ten days post seeding, the plates were washed with PBS and stained with crystal violet solution (Sigma Aldrich).

**Statistics**. To compare between treatment groups, a two-tailed Student $t$-test was used if not otherwise stated. Data analysis of DigiWest and expression data was performed in R. Differential DigiWest signal and gene expression were tested using the moderated t-test implemented in the 'limma' R/Bioconductor package[79]. Gene set enrichment tests were performed using the permutation test implemented in the 'fgsea' R/Bioconductor package[78]. Multiple testing correction with the Benjamini-Hochberg method was performed in all applicable cases. All tests were two-sided.

**Reporting summary**. Further information on research design is available in the Nature Research Reporting Summary linked to this article.

## Data availability

Computer code to reproduce all gene expression analyses presented in this study is available at https://github.com/boutroslab/Supplemental-Material. Expression data of microarrays is accessible through GEO under the accession number GSE114061.

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

## Acknowledgements

We are grateful to F. Heigwer, L. Henkel, J. Winter, D. Kranz, O. Voloshanenko, S. Acebrón and F. Port for helpful comments on the manuscript and valuable discussion. We thank D. Brügemann, I. Yildiz, C. Dingert, A. Falzone and S. Oettinger for experimental assistance. We thank H. Farin for valuable advices on organoid culture. We thank the DKFZ Genomics and Proteomics Core Facility for helping with the Illumina Expression analysis. B.R. was supported by the BMBF-funded Heidelberg Center for Human Bioinformatics (HD-HuB) within the German Network for Bioinformatics Infrastructure (de.NBI). J.B. was supported by the "Translational Physician Scientist (TRAPS)" program of the Medical Faculty Mannheim, Heidelberg University and the State of Baden-Württemberg. M.P.E was supported by grants from the State of Baden-Württemberg for the "Center of Geriatric Oncology (ZOBEL) – Perspektivförderung" and "Biology of Frailty – Sonderlinie Medizin". The groups of M.B. and M.P.E. were supported by the Hector Foundation II.

## Author contributions

T.Z. and M.B. conceived and planned the experiments. T.Z., G.A., M.W., J.B., N.R., I.H., R.H., L.B., G.E. and B.H. performed the experiments; B.R. performed data analysis; E.B., I.H. and K. M.-D. conceived and performed mouse and xenograft experiments; T.Z. and M.B. wrote the manuscript, M.P.E. and M.B. supervised the project and acquired funding

## Additional information

**Competing interests:** The authors declare no competing interests.

