## [Peer Review File · Nature Communications]

Reviewers' comments:

Reviewer #1 (Remarks to the Author):

In this report, the authors describe an intriguing observation that inhibition of MEK, either chemically or gene silencing, leads to the reduction in Axin1 level and increases in Wnt- β -catenin reporter gene activity and Wnt target gene expression in colorectal cancer cells lacking APC. The authors provide ample evidence to support this conclusion and demonstrate that these effects of MEK inhibition depend on the lack of APC. However, the manuscript falls short in characterization of the mechanism by which MEK inhibition results in Wnt- β -catenin activation and does not provide any evaluation of the inhibition in tumor cell growth in culture and tumor formation or progression in vivo. The lack of the latter significantly compromises the potential significance of the findings.

- 1) The three cell lines used in the initial screen all harbor KRAS mutations. However, the RAF inhibitor was described not effective. Authors should at least discuss this
- 2) GSK3 inhibition in APC null cells has no effect on Axin2 expression. However, in Fig. 5C GSK3i also reduced Axin1 levels (actually more than MEKi). Does this suggest that AXIN1 reduction may not be the major mechanism for MEKi-induced Wnt activity? Along the same line, if GSK3 inhibition has no effect on Wnt activity in APC-null cells, would its dissociation from AXIN1 be important for increasing Wnt activity in these APC-null cells?
- 3) There appears to be an increase in Axin2 expression of WT colon of mice treated with MEKi. Does this suggest that mouse colon and small intestinal cells behave differently (colon cell can respond to MEKi in the presence of APC)?
- 4) To better understand the human relevancy of this study, the human sample analysis needs to be accompanied by the knowledge of the APC and KRAS genotype of the samples.
- 5) The major shortfall of this study is the lack of biological characterization of MEKi. In the APC-null cells, the Wnt- β -catenin transcriptional activity is very high. It is very important to know whether further increases in the activity by MEKi can be translated into biological outcomes (cell proliferation, organoid formation, tumor progression, etc).
- 6) MEKi has been tested in a clinical trial as mentioned in the manuscript. If the authors' findings are biological significant, should there be evidence for MEKi treatment to negatively impact CRC progression from the trial?

Reviewer #2 (Remarks to the Author):

The manuscript by Zhan et al. describes the identification of MEK1/2 inhibitors as activators of Wnt/b-catenin signaling in CRC cells, and how this effect is mediated by the AXIN1 downregulation by transcriptional repression, leading to enrichment of gene signatures associated with stemness, suggesting that MEK1/2 might induce stem cell plasticity and therefore resistance in CRC. Although the interaction between the Wnt and BRAF pathways is not novel, and in fact, a similar upregulation of the Wnt pathway upon inhibition of the EGFR/RAS/RAF/MEK/ERK pathway in CRC was described recently by Chen et al, *Mol Cancer Therapeutics* 2018 (reference should be included), the data in this manuscript seems to suggest that the MEK1/2 and ERK1/2 inhibition produces its effects on CRC stemness by AXIN1 downregulation, which has not been described before. In general, Zhan et al. do an extensive study on the effects of MEK1/2 inhibitors on Wnt signal regulation and stemness, but I believe that to merit publication in *Nature Communication*, this work needs to include data on how possible combinations of Wnt pathway and MEK1/2 inhibitors (eg, trametinib and PRI-724, as the authors suggest) are more efficacious in vitro using cell viability assays, and have higher efficacy in CRC tumor reduction and survival in CRC animal models, preferentially PDXs. It is also intriguing that in general, EGFR and BRAF inhibitors have no or very minor effects on activating the Wnt pathway on the CRC cells, respectively (Fig 1C, D), suggesting that the effect seen by MEK1/2 and ERK1/2 inhibitors it is not by inhibiting the RAS pathway, but perhaps alternative more complex processes involving transcription regulation effects on transcription factors and super-enhancers that regulate cell plasticity, which have been described before (Zawistowski et al. *Cancer Discov* 2017, 7(3): 302–321; Yohe et al., *Science Trans Med* 2018, Jul 4;10(448)). Given that the authors demonstrate that MEK1/2 inhibitors downregulate AXIN1 at the transcriptional levels, they should include data to investigate whether this effect is due to remodeling of super-enhancers or other transcriptional effects that link Mek1/2 inhibition to AXIN1 transcription.

Additional comments:

- The doses of MEK1/2 chosen for the studies shown in Fig 2 are 10 or 100-fold higher than the cell-free based IC50 for those MEK1/2 inhibitors, so it is difficult to assess the potency of the compounds in these cell-based assays; data for Fig 2 A, B, E should be presented with a lower range of compound doses. Similar for figures Fig. 4C and D, Fig 5D. Interestingly, a more appropriate range of doses of MEKi was used for figures 6D and E and 7F, showing nice titration of the signal.
- In page 10, a reference should be added to the statement: “ In previous mass spectrometry experiments, ...”
- Given that ERK1/2 inhibitors also have an effect upregulating Wnt in CRC cells, albeit less potent than MEK1/2 inhibitors, it would be interesting to show whether they also downregulate AXIN1 levels as MEK1/2 inhibitors do. In this regard, the authors should test a couple of additional ERK1/2 inhibitors (eg, SCH772984 and LY3214996) of similar cell-free IC50 to confirm that the effect is on target
- In page 14, and other instances, the authors state “We propose that the reprogramming of cancer cells is a yet unknown side effect of RAS pathway inhibition” given that only MEK1/2 and

ERK1/2 at a lesser extent, but not EGFR and BRAF inh, have an effect on the proposed reprogramming mechanism, it does not seem that this statement is accurate. In the same page, there is a discussion on feedback loops within the RAS pathway depending on the inhibition point, and the authors also state that “cellular process which are activated upon interference with MEK are more complex”. This last statement is actually quite critical: why are the effects different by inhibiting the RAS pathway upstream vs MEK or ERK? See comments above regarding additional studies

- There are two sections on the generation of Wnt reporter cell lines in the “ONLINE METHODS”, in pages 19 and 22. What data was generated by each one of the methods? On page 19, was the screen implemented using pools of transiently transfected cells or isolated clonal cells? On page 22, it appears like the cells were transfected with two separate plasmids into cells seeded in wells of a 384-well plate; this protocol it is not very smart if the cells are to use for screening because you increase variability; transduction in flasks and the seeding in wells is the way to do it. But then, it is not clear what cells transduced with this protocol were used for.
- For the “Small molecule screen” protocol (page 20), I’m assuming there were two sets of parallel assay plates, one for the luciferase and another for the Celltiterglo readouts. Please clarify and explicitly indicate that this is indeed the case so there is no confusion to the reader; All the data for each readout (as % activity of DMSO control) from the screens done should be made available to the reader as supplemental material. The Wnt luc data normalized to cell number by dividing by the CellTiterGlo can be misleading if the number of cells is small due to cell cytotoxicity, especially at a dose of 10 uM compound with a focused collection, which likely includes a lot of “cancer” compounds. As a supplemental, the authors should include a heat map of % activity for each compound with each separate readout eg, wnt luc and Celltiterglo.

Response to Reviewers

We thank the reviewer for the helpful comments and have addressed them by performing additional experiments and clarifications in the text (changes are colored in red in the manuscript text).

Specifically, we have

1. Performed experiment to demonstrate that only MEK inhibitors, but not inhibitors of EGFR or RAF, are sufficient to inhibit the Ras pathway at the level of phospho-ERK and thus activate Wnt (new Figure S2B).
2. Added novel experiments that demonstrate that AXIN1 is rapidly lost upon incubation with MEK inhibitors, which results in an induction of Wnt target gene expression. We updated figures (new Figure 5 and new Figure S6) and rewrote the manuscript text (pages 9-10,11-12,17) to further clarify the role of AXIN1 in MEK inhibitor induced Wnt activation.
3. Identified EGR1 as a transcriptional factor that links Ras signaling with transcriptional controls of AXIN1 expression (new Figure S7, pages 10-11, 17).
4. Confirmed the activating effect of ERK inhibition on Wnt signaling using a second small molecule inhibitor of ERK1/2 (new Figure 3D) and Performed *in vitro* combinatorial treatments with MEK and Wnt inhibitors and demonstrated an enhanced antiproliferative effect of combined inhibition of both pathways (new Figure S9A).
5. Established a novel xenograft model using patient-derived colorectal cancer organoids. Using this model, we show that combined pharmacological inhibition of MEK and Wnt is superior in reducing tumor growth than single treatments (new Figure S9B, pages 15, 19).
6. Improved presentation of our drug screens by adding heatmaps of each experiment (new Figure S1B, S2A) and providing all results as data tables (New Table S1 and S2). We have also rewritten the Methods section to clarify our experimental approach (page 21-22).

A point by point response is appended below.

Reviewer #1:

“In this report, the authors describe an intriguing observation that inhibition of MEK, either chemically or gene silencing, leads to the reduction in Axin1 level and increases in Wnt- β -catenin reporter gene activity and Wnt target gene expression in colorectal cancer cells lacking APC. The authors provide ample evidence to support this conclusion and demonstrate that these effects of MEK inhibition depend on the lack of APC.”

We thank the reviewer for the overall positive comment on our work. We would like to emphasize that APC truncations are not obligatory for MEK inhibitors to induce Wnt signaling, as for example HCT116 cells only harbor an activating mutation of CTNNB1. However, we provide experimental data which shows that APC truncations can strongly enhance the Wnt activating effect of MEK inhibitors in colorectal cancer cell lines (Figure 4). We have rewritten the manuscript to present this observation more clearly (page 8).

“However, the manuscript falls short in characterization of the mechanism by which MEK inhibition results in Wnt- β -catenin activation and does not provide any evaluation of the inhibition in tumor cell growth in culture and tumor formation or progression in vivo. The lack of the latter significantly compromises the potential significance of the findings.”

To address this comment, we have performed additional experiments which have been added to the manuscript. We, however, respectfully disagree with the statement that we have not characterized the mechanism and would like to iterate what we have done. First, we performed a high-throughput Western Blot experiment (DigiWest) to identify genes in the Wnt and Ras pathway that are differentially regulated on a protein level upon MEK inhibition (Figure 5A). With this method, we identified a downregulation of AXIN1, which we could confirm in different colon cancer cell lines (Figure 5B, Figure S6A) and human colorectal cancer organoids (Figure 7D). We then show that AXIN1 protein and transcript levels are rapidly depleted upon MEK inhibition (new Figure 5C). To further confirm the role of AXIN1 in MEK inhibitor induced Wnt activation, we depleted AXIN1 by siRNA and could demonstrate an induction of the Wnt target gene AXIN2 upon knockdown (Figure 5E). In line with this observation, stabilization of AXIN by tankyrase inhibitors could abolish the stimulating effect of MEK inhibitors on Wnt signaling (Figure S6E). We have rewritten the manuscript to present our results more clearly (pages 9-10, 11-12, 17).

Furthermore, we added novel experimental data showing that EGR1, a downstream target of Ras signaling and known transcriptional activator of AXIN1 expression¹, is rapidly downregulated upon MEK inhibition in both cancer cell line and colon cancer organoids (new Figure S7A-D). Furthermore, downregulation of AXIN1 expression could be rescued by overexpression of EGR1 (new Figure S7E). These data suggest that EGR1 is a functional link between Ras activity and AXIN1 levels (see also pages 10-11, 17). In our opinion, these experimental data provide novel insights into the underlying mechanism of Wnt activation caused by MEK inhibition.

Regarding the effect on cancer growth phenotypes, we now provide experimental data showing a superior effect of combinatorial Wnt and MEK inhibition in a colon cancer cell line (new Figure S9A) and *in vivo* in a patient derived xenograft model using colorectal cancer organoids (new Figure S9B). These data support our thesis that Wnt activation is a salvage pathway that alleviates the effect of MEK inhibitors.

“1) The three cell lines used in the initial screen all harbor KRAS mutations. However, the RAF inhibitor was described not effective. Authors should at least discuss this.”

We would like to thank the reviewer for pointing this out. Our experiments indicate that an efficient inhibition of the Ras-MAPK pathway, resulting in a reduction of phospho-ERK, is necessary to activate Wnt signaling in Ras mutant colorectal cancer (Figure 3E). In contrast to MEK inhibitors, small molecules that inhibit the RAF kinase can re-activate the Ras pathway through c-RAF and this feedback loop is specific for KRAS mutant colorectal cancer, as previously shown². Thus, we assume that the inability of RAF inhibitors to block the Ras pathway at the level of phospho-ERK is the reason for their lacking effect on Wnt signaling. To experimentally support this notion, we performed additional experiments and treated KRAS mutant colorectal cancer cell lines with a panel of EGFR (erlotinib, gefitinib) and RAF (RAF265, dabrafenib) inhibitors and compared their effect to trametinib and GDC-0994. We show that phospho-ERK and phospho-RSK levels were only abolished by MEK and ERK inhibitors, but not by small molecules targeting EGFR or RAF, underlining that shut-down of the Ras pathway is required to activate Wnt signaling (new Figure S2B). A detailed explanation has been added to page 6.

“2) GSK3 inhibition in APC null cells has no effect on Axin2 expression. However, in Fig. 5C GSK3i also reduced Axin1 levels (actually more than MEKi). Does this suggest that AXIN1 reduction may not be the major mechanism for MEKi-induced Wnt activity? Along the same line, if GSK3 inhibition has no effect on Wnt activity in APC-null cells, would its dissociation from AXIN1 be important for increasing Wnt activity in these APC-null cells?”

We thank the reviewer for the comment and would like to clarify how we demonstrated that reduction of AXIN1 is a mechanism by which MEK inhibitors stimulate Wnt signaling. First, we observed in both SW480 and HCT116 (Figure 5B, Figure S6A) as well as in human colorectal cancer organoids (Figure 7D) that AXIN1 levels are downregulated upon MEK inhibition. For HCT116, we show that siRNA mediated knockdown of AXIN1 can activate Wnt signaling by inducing AXIN2 expression (Figure 5E), whereas stabilization of AXIN by tankyrase inhibitors can block MEK inhibitor induced Wnt activation (Figure S6E). We believe that these data clearly show that modifying levels of AXIN1 has an impact on Wnt signaling.

As pointed out by the reviewer, we also observed that GSK3B inhibitors caused a similar reduction of AXIN1 levels in SW480 and HCT116 (Figure 5B, Figure S6A). In both cell lines, GSK3B inhibitors also activate Wnt signaling as shown in our focused kinase inhibitor screen (Figure 1D) and by single experiments in SW480 (Figure 2A - BIO) or in HCT116 (Figure 2E). We reviewed the literature and did not find any studies that described downregulation of AXIN1 upon GSK3B inhibition in colorectal cancer. Although this observation is interesting, we did not further analyze the underlying mechanisms as we mainly focused on MEK inhibitors. However, we do not see any contradictions of this observation with our proposed role of AXIN1 in activating the Wnt pathway.

Beside the reduction of AXIN1, our experimental data suggest that MEK inhibition activates additional mechanisms leading to enhanced Wnt signaling. We propose that one mechanism is the dissociation of AXIN1 from GSK3B as was observed in SW480 (Figure 5F). This change in protein interaction at the level of the destruction complex was shown to take place during Wnt activation³. We assume that the genetic background of the tumor cells determine the contribution of either mechanism (AXIN1 downregulation or dissociation of AXIN1 from GSK3B) to MEK inhibitor induced Wnt activation.

“3) There appears to be an increase in Axin2 expression of WT colon of mice treated with MEKi. Does this suggest that mouse colon and small intestinal cells behave differently (colon cell can respond to MEKi in the presence of APC)?”

The reviewer correctly noted that expression of specific stem cell markers such as LGR5 is increased exclusively in the small intestine upon trametinib treatment. In contrast, elevated AXIN2 expression was only observed in the colon after MEK inhibition. We believe that this difference is the result of context-dependent Wnt responses. First, it is known that target genes of the Wnt pathway can differ between tissue types, including small intestine and colon⁴ and even between tumors of the same tissue origin⁵. Secondly, the basal level of Wnt activity is different in both tissue compartments. For instance, mice that harbor heterozygous mutations at different positions of APC either develop polyps in the small intestine or the colon, indicating that different levels of Wnt are required to initiate tumorigenesis in specific parts of the digestive system⁶. Furthermore, studies that established organoid culture of the gastrointestinal tract demonstrated that murine colon organoids depend on supplementation with Wnt ligands for survival⁷ whereas small intestinal organoids have a much higher basal level of Wnt activity and grow independently of external Wnt ligands⁸. Due to this difference in basal Wnt activity, we believe that it is not surprising that the effect of MEK inhibitors on Wnt signaling and expression of Wnt target genes will also differ between colon and small intestine.

“4) To better understand the human relevancy of this study, the human sample analysis needs to be accompanied by the knowledge of the APC and KRAS genotype of the samples.”

We fully agree and performed amplicon sequencing of 46 genes captured with 157 amplicons. This panel of amplicons was designed to cover the most frequently altered hotspot locations in colorectal cancer. The genotypes of the organoids are presented in the Table S4 and we added a section to the results to underline this aspect (page 13).

“5) The major shortfall of this study is the lack of biological characterization of MEKi. In the APC-null cells, the Wnt- β -catenin transcriptional activity is very high. It is very important to know whether further increases in the activity by MEKi can be translated into biological outcomes (cell proliferation, organoid formation, tumor progression, etc).”

The reviewer correctly points out that it is important to understand the biological impact of Wnt activation resulting from MEK inhibition, particularly in cell lines that already harbor a mutation in the Wnt pathway (e.g. APC in RKO cells). We fully agree with her/his point but argue that high Wnt activation does not necessarily need to translate into a growth or proliferation phenotype *in vitro*, especially in the presence of a Ras pathway inhibitor. Rather, based on our experiments with patient-derived cancer organoid, we demonstrated that MEK inhibition leads to transcriptome changes that are characteristic for intestinal stem cells and predict colorectal cancer which have a high risk of relapse⁹ (Figure 8A-B). We do not believe that this phenotype can be easily recaptured by *in vitro* experiments. Furthermore, we observed that MEK inhibition leads to a reduced growth rate of colorectal cancer cell lines while increasing their Wnt activity. This combination of characteristics is found in quiescent cancer stem cells, as shown by previous studies^{10,11}. We assume that this cellular state facilitates survival of cells during Ras pathway inhibition. To validate this hypothesis, we performed additional experiments with combined inhibition of both Ras and Wnt pathway in colorectal cancer cell line and PDX models. We demonstrate that blocking Wnt signaling acts synergistically with MEK inhibition to reduce cell proliferation (new Figure S9A, S9B). This result suggests that Wnt activation is a salvage mechanism that rescues the anti-proliferative effect of MEK inhibition. These novel findings were added to the manuscript on page 15.

“6) MEKi has been tested in a clinical trial as mentioned in the manuscript. If the authors' findings are biological significant, should there be evidence for MEKi treatment to negatively impact CRC progression from the trial?”

A number of phase I-II clinical trials were conducted to test the activity of different MEK inhibitors in solid cancers, including colorectal cancer^{12,13}. Results of these trials demonstrated that MEK inhibitors could not reduce the growth of colorectal cancer in spite of detectable reduction in phospho-ERK levels. In contrast, patients with melanoma benefited significantly from MEK inhibitor treatment¹⁴. As a consequence, further investigations of MEK inhibitors as monotherapy in colorectal cancer were discontinued. We changed the manuscript to present this aspect more clearly (page 16). The mechanisms underlying the resistance of colorectal cancer to MEK inhibition was not understood. It was proposed that the Ras pathway was not sufficiently blocked by MEK inhibitors¹⁵. However, targeting of the Ras pathway by combined MEK, EGFR and RAF inhibition in BRAF mutant colorectal cancer only resulted in a 20% response rate¹⁶, indicating that additional cellular mechanisms acting outside the Ras pathway contribute to drug resistance.

Reviewer #2:

“The manuscript by Zhan et al. describes the identification of MEK1/2 inhibitors as activators of Wnt/ β -catenin signaling in CRC cells, and how this effect is mediated by the AXIN1

downregulation by transcriptional repression, leading to enrichment of gene signatures associated with stemness, suggesting that MEK1/2 might induce stem cell plasticity and therefore resistance in CRC. Although the interaction between the Wnt and BRAF pathways is not novel, and in fact, a similar upregulation of the Wnt pathway upon inhibition of the EGFR/RAS/RAF/MEK/ERK pathway in CRC was described recently by Chen et al, Mol Cancer Therapeutics 2018 (reference should be included), the data in this manuscript seems to suggest that the MEK1/2 and ERK1/2 inhibition produces its effects on CRC stemness by AXIN1 downregulation, which has not been described before.”

We thank the reviewer for the positive comment but respectfully disagree that our results overlap with the observation made by Chen et al. In the mentioned study, it was shown that BRAF inhibitors could activate Wnt signaling in BRAF mutant colorectal cancer. While this observation points to a similar direction as our findings, we did not observe any Wnt stimulating effects for most BRAF or c-RAF inhibitors in our KRAS mutant colorectal cell lines (Figure 1C). This difference in response between MEK and RAF inhibition indicates that distinct mechanisms of Wnt activation exist which depend on the specific activating mutation of the Ras pathway. Similarly, BRAF inhibitors were shown to activate Wnt signaling in BRAF mutant melanoma¹⁷. However, Biechele et al. demonstrated that Wnt activity is required in melanoma to induce apoptosis, which indicates an opposite biological effect of Wnt activation compared to colorectal cancer. Therefore, we believe it is important to investigate the effect of Wnt signaling in a tissue and mutation-specific context.

Additionally, we put significant efforts to investigate the underlying mechanisms of Wnt activation. We used different experimental methods and cancer cell models to demonstrate a critical role of AXIN1 as a mediator of Wnt activation (new Figure 5, Figure S6). Therefore, we believe that our study delivers novel insights that extend beyond the work of Chen et al.

“1) I believe that to merit publication in Nature Communication, this work needs to include data on how possible combinations of Wnt pathway and MEK1/2 inhibitors (eg, trametinib and PRI-724, as the authors suggest) are more efficacious in vitro using cell viability assays, and have higher efficacy in CRC tumor reduction and survival in CRC animal models, preferentially PDXs.”

We thank the reviewer for his/her comment and performed additional *in vitro* and *in vivo* experiments using combinatorial treatments with MEK and Wnt inhibitors, as suggested. We could demonstrate that combined inhibition of the Wnt and Ras pathway using trametinib and PRI-724 resulted in a synergistic reduction of cell survival in HCT116 cells *in vitro* (new Figure S9A). In addition, we developed a PDX model by engraftment of colorectal cancer organoids. In this model, we show that combinatorial treatment with trametinib and ICG-001 was superior in reducing tumor growth than single agent treatments (new Figure S9B). Both results indicate that targeting Wnt and MEK in parallel could be a promising therapeutic approach for colorectal cancer and are now described on pages 15 and 19.

“2) It is also intriguing that in general, EGFR and BRAF inhibitors have no or very minor effects on activating the Wnt pathway on the CRC cells, respectively (Fig 1C, D), suggesting that the effect seen by MEK1/2 and ERK1/2 inhibitors it is not by inhibiting the RAS pathway, but perhaps alternative more complex processes involving transcription regulation effects on transcription factors and super-enhancers that regulate cell plasticity, which have been described before (Zawistowski et al. Cancer Discov 2017, 7(3): 302–321; Yohe et al., Science Trans Med 2018, Jul 4;10(448)).”

The reviewer correctly pointed out that EGFR and BRAF inhibitors did not activate Wnt signaling in our kinase inhibitor screen, as opposed to MEK inhibitors (Figure 1C-D). We observed that Wnt activation occurs only if the Ras-MAPK pathway is sufficiently inhibited at

the level of phospho-ERK or -RSK (Figure 3E). As previously shown by others, targeting Ras at the level of EGFR is insufficient to completely block downstream signaling in the presence of KRAS mutations¹⁸. Hence, RAS mutations are well established clinical markers that predict individuals with colorectal cancer who will not benefit from treatment with anti-EGFR antibodies¹⁹. Likewise, small molecule inhibitors of RAF can reactivate Ras signaling by acting through c-RAF². Therefore, both RAF and EGFR inhibitors are unable to block Ras signaling and therefore activate Wnt. To support this assumption, we performed additional experiments and compared the effect of different EGFR, RAF and MEK inhibitors on phospho-ERK and phospho-RSK levels. We show that only MEK and ERK inhibitors can block the Ras pathway in KRAS mutant colorectal cancer cell lines (new Figure S2B). These results are not presented in the manuscript text on page 6.

“3) Given that the authors demonstrate that MEK1/2 inhibitors downregulate AXIN1 at the transcriptional levels, they should include data to investigate whether this effect is due to remodeling of super-enhancers or other transcriptional effects that link Mek1/2 inhibition to AXIN1 transcription.”

We performed a series of additional experiments to further understand the mechanism by which AXIN1 levels are downregulated by MEK inhibitors.

First, we performed kinetic assays to investigate how fast AXIN1 levels are reduced upon MEK inhibitor treatment. The results show that within two hours, cytoplasmic levels of AXIN1 were significantly depleted (new Figure 5C). In parallel, we observed that the expression of the Wnt target gene AXIN2 is induced, which is accompanied by a repression of AXIN1 transcript levels (new Figure 5D). Based on our DigiWest results, we identified EGR1 as a transcription factor that is strongly depleted upon MEK inhibitor treatment (Figure S7A). In kinetics assays, we observed that EGR1 protein and transcript levels are rapidly reduced within two hours after treatment with trametinib (new Figure S7B-C). EGR1 is a downstream target of the Ras pathway^{20,21} and binds to the enhancer region of AXIN1, causing its transcriptional activation¹. We show that EGR1 is significantly downregulated not only in cancer cell lines, but also in different colorectal cancer organoid lines upon trametinib treatment (new Figure S7D). Furthermore, overexpression of EGR1 could prevent downregulation of AXIN1 transcript levels by trametinib treatment (new Figure S7E). Therefore, these novel experimental results indicate that EGR1 could be a mechanistic link between Ras pathway activity and transcriptional regulation of AXIN1 gene expression. These findings are now described on page 10-11, 17.

“4) The doses of MEK1/2 chosen for the studies shown in Fig 2 are 10 or 100-fold higher than the cell-free based IC₅₀ for those MEK1/2 inhibitors, so it is difficult to assess the potency of the compounds in these cell-based assays; data for Fig 2 A, B, E should be presented with a lower range of compound doses. Similar for figures Fig. 4C and D, Fig 5D. Interestingly, a more appropriate range of doses of MEKi was used for figures 6D and E and 7F, showing nice titration of the signal.”

It is correct that the cell free IC₅₀ of trametinib is ranging from 0.94 nM to 3.4 nM in cell free assays that used active BRAF and c-RAF proteins to stimulate MEK and ERK phosphorylation²². However, in cell culture, levels of trametinib that are necessary to achieve complete inhibition of phospho-ERK levels and growth arrest were much higher, particularly in KRAS mutant colorectal cancer cells, as previously shown²². At a concentration of 100 nM, it was shown that trametinib could block ERK phosphorylation in nearly all colorectal cancer cell lines²². This is also supported by our dose response analysis of Wnt activation by trametinib (Figure 2C). At a concentration of 100 nM, we could observe activation of Wnt by the MEK inhibitor. We also believe that our observation is not caused by off-targets of the compounds, as trametinib is considered to be a very specific inhibitor²². Furthermore, we could confirm our results by an independent method, namely siRNA mediated depletion of MEK1/2 (Figure 3A).

These separate methods underline that the effect on Wnt activation is specifically caused by MEK inhibition.

“5) In page 10, a reference should be added to the statement: ‘In previous mass spectrometry experiments, ...’”

The mentioned mass spectrometry experiments were performed in our group. Based on these results, we performed further co-immunoprecipitation experiments that confirmed a dissociation of AXIN1 from GSK3 (Figure 5F). We changed the manuscript to clarify this point (page 11).

“6) Given that ERK1/2 inhibitors also have an effect upregulating Wnt in CRC cells, albeit less potent than MEK1/2 inhibitors, it would be interesting to show whether they also downregulate AXIN1 levels as MEK1/2 inhibitors do. IN this regard, the authors should test a couple of additional ERK1/2 inhibitors (eg, SCH772984 and LY3214996) of similar cell-free IC50 to confirm that the effect is on target”

We performed additional experiments to show that ERK1/2 inhibition also reduces AXIN1 levels, albeit to a lesser degree than MEK inhibitors (new Figure S6D). Additionally, we tested the effect of a second ERK1/2 Inhibitor, SCH772984, and could observe a similar stimulating effect on AXIN2 expression in both HCT116 and SW480 cells (new Figure 3D). Since we also show that the Wnt pathway is activated upon siRNA mediated knockdown of ERK1/2 (Figure 3B), we are convinced that the effect on Wnt signaling is specific for ERK1/2 inhibition. The results are described on page 7 and page 10.

“7) In page 14, and other instances, the authors state “We propose that the reprogramming of cancer cells is a yet unknown side effect of RAS pathway inhibition” given that only MEK1/2 and ERK1/2 at a lesser extent, but not EGFR and BRAF inh, have an effect on the proposed reprograming mechanism, it does not seem that this statement is accurate. In the same page, there is a discussion on feedback loops within the RAS pathway depending on the inhibition point, and the authors also state that “cellular process which are activated upon interference with MEK are more complex”. This last statement is actually quite critical: why are the effects different by inhibiting the RAS pathway upstream vs MEK or ERK? See comments above regarding additional studies.”

We would like to thank the reviewer for pointing this out. The effective blockage of the Ras pathway, which is commonly measured at the level of phospho-ERK, is highly dependent on the genetic (KRAS or BRAF mutation) and tissue background of the cancer cells. For colorectal cancer, there is strong experimental as well as clinical evidence that KRAS mutations renders the Ras pathway insensitive towards EGFR inhibitors such as cetuximab^{18,19,23}. Therefore, anti-EGFR antibodies are not recommended in patients with confirmed KRAS mutation. In the case of RAF inhibitors, which target the pathway downstream of KRAS, it was shown that the compounds can reactivate Ras signaling and ERK phosphorylation through c-RAF in KRAS mutant cancer². Thus, we assume that targeting EGFR or BRAF are not efficient enough to shut down the Ras pathway, which is the reason for their inability to activate Wnt signaling. To support this point, we treated KRAS mutant colorectal cancer cell lines (HCT116, SW480) with different EGFR and RAF inhibitors and compared their effect on phospho-ERK levels to MEK and ERK inhibitors. We could show that only MEK and ERK inhibitors could block phosphorylation of ERK and its downstream kinase RSK (new Figure S2B). Therefore, we believe that this is the reason why only MEK and ERK inhibitors can activate Wnt signaling in KRAS mutant colorectal cancer.

“8) There are two sections on the generation of Wnt reporter cell lines in the “ONLINE METHODS”, in pages 19 and 22. What data was generated by each one of the methods? On

page 19, was the screen implemented using pools of transiently transfected cells or isolated clonal cells? On page 22, it appears like the cells were transfected with two separate plasmids into cells seeded in wells of a 384-well plate; this protocol it is not very smart if the cells are to use for screening because you increase variability; transduction in flasks and the seeding in wells is the way to do it. But then, it is not clear what cells transduced with this protocol were used for.”

We have rewritten this section to clarify our experimental approach (page 21-22). For the screening experiments, we used cell lines that stably express the respective Wnt/TCF reporters. These reporter lines were generated by lentiviral infection of pools of colorectal cancer cells, followed by antibiotic selection. This approach was selected to ensure that the variability between the technical and biological replicates are minimized. However, since most Wnt reporter assays are based on dual expression of reporter plasmids (e.g. TOP/Flash), which yields a higher luciferase signal, we used this method to determine Wnt activity in the isogenic RKO cell lines, which have a low basal level of Wnt signaling, upon MEK inhibition (Figure 4D-E). We adapted our manuscript to clarify this point (page 5 and 8).

“9) For the “Small molecule screen” protocol (page 20), I’m assuming there were two sets of parallel assay plates, one for the luciferase and another for the Celltiterglo readouts. Please clarify and explicitly indicate that this is indeed the case so there is no confusion to the reader; All the data for each readout (as % activity of DMSO control) from the screens done should be made available to the reader as supplemental material. The Wnt luc data normalized to cell number by dividing by the CellTiterGlo can be misleading if the number of cells is small due to cell cytotoxicity, especially at a dose of 10 uM compound with a focused collection, which likely includes a lot of “cancer” compounds. As a supplemental, the authors should include a heat map of % activity for each compound with each separate readout eg, wnt luc and Celltiterglo”.

It is correct that we used two parallel sets of plates for the luciferase assay and for the CellTiterGlo readout. We have rewritten the text to clarify our experimental approach (page 21-22). Furthermore, we added all screening data as heatmaps (new Figure S1B and S2A) and data tables (new Table S1 and S2) to the supplement of our manuscript. All data were normalized to the DMSO controls of the respective assay plates. The reviewer correctly pointed out that cellular toxicity could be a potential cofounder in our approach, as we normalized Wnt reporter activity to cellular viability. However, as shown in the provided heatmaps and raw data, the majority of compounds did not cause a strong viability phenotype within the 24 h treatment period. Therefore, we assume that the rate of false positive or negative hits is low.

References:

1. Zhang, M., Liao, Y. & Lönnnerdal, B. EGR-1 is an active transcription factor in TGF- β 2-mediated small intestinal cell differentiation. *J. Nutr. Biochem.* **37**, 101–108 (2016).
2. Hatzivassiliou, G. *et al.* RAF inhibitors prime wild-type RAF to activate the MAPK pathway and enhance growth. *Nature* **464**, 431–435 (2010).
3. Liu, X., Rubin, J. S. & Kimmelman, A. R. Rapid, Wnt-induced changes in GSK3 β associations that regulate β -catenin stabilization are mediated by G α proteins. *Curr. Biol.* **15**, 1989–1997 (2005).
4. Ramakrishnan, A.-B. & Cadigan, K. M. Wnt target genes and where to find them. *F1000Research* **6**, 746 (2017).
5. Herbst, A. *et al.* Comprehensive analysis of β -catenin target genes in colorectal carcinoma cell lines with deregulated Wnt/ β -catenin signaling. *BMC Genomics* **15**, 74 (2014).
6. McCart, A. E., Vickaryous, N. K. & Silver, A. Apc mice: Models, modifiers and mutants. *Pathol. Res. Pract.* **204**, 479–490 (2008).
7. Sato, T. *et al.* Long-term expansion of epithelial organoids from human colon,

- adenoma, adenocarcinoma, and Barrett's epithelium. *Gastroenterology* **141**, 1762–1772 (2011).
8. Jung, P. *et al.* Isolation and in vitro expansion of human colonic stem cells. *Nat. Med.* **17**, 1225–7 (2011).
 9. Merlos-Suárez, A. *et al.* The intestinal stem cell signature identifies colorectal cancer stem cells and predicts disease relapse. *Cell Stem Cell* **8**, 511–24 (2011).
 10. Wei, L. *et al.* Inhibition of CDK4/6 protects against radiation-induced intestinal injury in mice. *J. Clin. Invest.* **126**, 4076–4087 (2016).
 11. Shimokawa, M. *et al.* Visualization and targeting of LGR5+ human colon cancer stem cells. *Nature* **545**, 187–192 (2017).
 12. Infante, J. R. *et al.* Safety, pharmacokinetic, pharmacodynamic, and efficacy data for the oral MEK inhibitor trametinib: A phase 1 dose-escalation trial. *Lancet Oncol.* **13**, 773–781 (2012).
 13. Rinehart, J. *et al.* Multicenter phase II study of the oral MEK inhibitor, CI-1040, in patients with advanced non-small-cell lung, breast, colon, and pancreatic cancer. *J. Clin. Oncol.* **22**, 4456–62 (2004).
 14. Falchook, G. S. *et al.* Activity of the oral MEK inhibitor trametinib in patients with advanced melanoma: a phase 1 dose-escalation trial. *Lancet Oncol.* **13**, 782–789 (2012).
 15. Corcoran, R. B. *et al.* Combined BRAF and MEK Inhibition With Dabrafenib and Trametinib in BRAF V600-Mutant Colorectal Cancer. *J. Clin. Oncol.* (2015). doi:10.1200/JCO.2015.63.2471
 16. Corcoran, R. B. *et al.* Combined BRAF, EGFR, and MEK Inhibition in Patients with BRAFV600E-Mutant Colorectal Cancer. *Cancer Discov.* **8**, CD-17-1226 (2018).
 17. Biechele, T. L. *et al.* Wnt/ β -catenin signaling and AXIN1 regulate apoptosis triggered by inhibition of the mutant kinase BRAFV600E in human melanoma. *Sci. Signal.* **5**, ra3-ra3 (2012).
 18. Misale, S. *et al.* Emergence of KRAS mutations and acquired resistance to anti-EGFR therapy in colorectal cancer. *Nature* **486**, 532–536 (2012).
 19. Lièvre, A. *et al.* KRAS mutation status is predictive of response to cetuximab therapy in colorectal cancer. *Cancer Res.* **66**, 3992–3995 (2006).
 20. Gregg, J. & Fraizer, G. Transcriptional Regulation of EGR1 by EGF and the ERK Signaling Pathway in Prostate Cancer Cells. *Genes Cancer* **2**, 900–909 (2011).
 21. Harada, T., Morooka, T., Ogawa, S. & Nishida, E. ERK induces p35, a neuron-specific activator of Cdk5, through induction of Egr1. *Nat. Cell Biol.* **3**, 453–459 (2001).
 22. Yamaguchi, T., Kakefuda, R., Tajima, N., Sowa, Y. & Sakai, T. Antitumor activities of JTP-74057 (GSK1120212), a novel MEK1/2 inhibitor, on colorectal cancer cell lines in vitro and in vivo. *Int. J. Oncol.* **39**, 23–31 (2011).
 23. Karapetis, C. S. *et al.* K-ras Mutations and Benefit from Cetuximab in Advanced Colorectal Cancer. *N. Engl. J. Med.* **359**, 1757–1765 (2008).

REVIEWERS' COMMENTS:

Reviewer #1 (Remarks to the Author):

The revision has addressed all of my concerns.

Reviewer #2 (Remarks to the Author):

Thank you to the authors for addressing the concerns; no additional comments from this reviewer.

AUTHORS' COMMENTS:

- We would like to thank both reviewers for their effort and positive comments on our revised manuscript.